# Interpreting biologically informed neural networks for enhanced proteomic biomarker discovery and pathway analysis

Erik Hartman [1,3] ✉, Aaron M. Scott [1,3], Christofer Karlsson [1], Tirthankar Mohanty [1], Suvi T. Vaara[2], Adam Linder [1], Lars Malmström [1] & Johan Malmström [1] ✉

The incorporation of machine learning methods into proteomics workflows improves the identification of disease-relevant biomarkers and biological pathways. However, machine learning models, such as deep neural networks, typically suffer from lack of interpretability. Here, we present a deep learning approach to combine biological pathway analysis and biomarker identification to increase the interpretability of proteomics experiments. Our approach integrates a priori knowledge of the relationships between proteins and biological pathways and biological processes into sparse neural networks to create biologically informed neural networks. We employ these networks to differentiate between clinical subphenotypes of septic acute kidney injury and COVID-19, as well as acute respiratory distress syndrome of different aetiologies. To gain biological insight into the complex syndromes, we utilize feature attribution-methods to introspect the networks for the identification of proteins and pathways important for distinguishing between subtypes. The algorithms are implemented in a freely available open source Python-package (https://github.com/InfectionMedicineProteomics/BINN).

The continuous technological advancements in mass spectrometry-based proteomics have enabled the quantification of hundreds to thousands of proteins in clinical samples extending its reach in biomedical and clinical research[1,2]. The increasing ability to rapidly analyze a large number of clinical samples provides new opportunities to profile complex biological systems and bridge the gap between translational and clinical research through the investigation of disease mechanisms and the identification of biomarkers. These advances are of interest for many disease areas, such as the study of infectious diseases where the identification of distinct clinical and molecular subphenotypes may impact the development of new treatment regimes. Subphenotypes are typically identified using clinical parameters based on the presented severity of different symptoms of the disease and are difficult to distinguish. Previous work has proposed

clinical subphenotypes for COVID-19[3–6] and sepsis[7–9], but the development of targeted treatments for the different subphenotypes remains challenging as the underpinning molecular mechanisms are poorly characterized. To understand these molecular mechanisms, it is therefore critical to analyze the proteins and associated biological pathways of a disease to support the development of precision treatments and provide the best patient care possible.

Currently, a common strategy to identify candidate diagnostic and prognostic biomarkers is based on significantly differentially expressed (DE) proteins between subphenotypes. Substantial research has been conducted on how to optimize DE detection algorithms[10–14], but the process of selecting proteins for further investigation remains unstandardized. In most cases, proteins that pass a *p*-value and fold-change threshold are considered the most informative, but these

[1]Division of Infection Medicine, Department of Clinical Sciences Lund, Faculty of Medicine, Lund University, Lund, Sweden. [2]Department of Perioperative and Intensive Care, University of Helsinki and Helsinki University Hospital, Helsinki, Finland. [3]These authors contributed equally: Erik Hartman, Aaron M. Scott. ✉e-mail: erik.hartman@hotmail.com; johan.malmstrom@med.lu.se

thresholds are rule based and potentially eliminate important biological signal. To understand the systemic impact of DE proteins, it is also pertinent to identify which pathways are enriched based on the difference in abundance of the DE proteins. Several tools and databases have been developed to automate this process and to select the most significant pathways based on the proteins that have been identified in DE analysis[15–17]. Commonly, the significance of a pathway is determined by counting the number of DE proteins that are connected to the pathway in a database and calculating a *p*-value based on these connections. This type of analysis typically omits crucial information such as protein abundance, protein co-expression, and pathway co-regulation, and selects the most interesting pathways using *p*-value cut-offs.

To mitigate these limitations, increasing efforts have been directed towards incorporating machine learning methods into proteomics workflows to improve the study of disease mechanisms and biomarker discovery[18–20]. Recent advances in the field of machine learning have allowed deep neural networks to thrive in domains of high dimensionality where complex networks can learn representations of features without the need for feature selection algorithms[21]. However, complex machine learning models, such as deep neural networks, suffer from a lack of interpretability, and although they provide greater predictive power than their more interpretable linear counterparts, this questions the utility of such methods. Research in the field of explainable artificial intelligence (xAI) has resulted in methods which allow for the interpretation of complex models by calculating the importance of each feature to the output of the model[22–24]. To further improve interpretability, biologically informed neural networks (BINNs) establish connections between their layers based on biological processes[25, 26] and thus generalize to unseen data more effectively[27].

Here, we demonstrate the utility of BINNs to develop highly accurate predictive models that enhance blood-based proteomics biomarker identification while providing greater insight into the underlying biology of a system. Using proteomic data as input, we annotate, train, and interpret BINNs in order to analyze the proteomic differences in blood plasma between subphenotypes of sepsis-induced acute kidney injury (AKI) and COVID-19. Through the interpretation of the trained BINNs, we identify panels of potential protein biomarkers that can stratify the AKI and COVID-19 subphenotypes with high accuracy and help provide a molecular explanation for the physical manifestation of the defined clinical subphenotypes. We also demonstrate how BINNs can be used for intelligent pathway analysis to extract the most important pathways in a biological system. To demonstrate the ability of the BINNs to generalize to different proteomics platforms, we utilized proteomics data generated by the Olink-platform to analyze the differences between various aeteologies of acute respiratory distress syndrome (ARDS). Overall, the inherent interpretability of BINNs lend to their potential to investigate complex biological systems in a more comprehensive manner and to enhance the potential of biomarker discovery in proteomics. A generalizable and user friendly software package for the creation and analysis of annotated sparse BINNs is open source and freely available at https://github.com/InfectionMedicineProteomics/BINN[28].

## Results

Currently, common proteomics-based biomarker identification and biological pathway analyses are based on thresholds which can omit important relationships in datasets, and therefore lack the comprehensiveness which is required when analyzing complex biological systems. Here, we apply a deep learning-based methodology which utilizes the Reactome pathway database[16] to incorporate biological relationships in a biologically informed neural network (BINN), allowing for a unified analysis of biomarkers, biological pathways, and biological processes. The Reactome database contains information about relationships of biological entities, and its underlying graph is manipulated to fit a sequential neural network-like structure, resulting

in a sparse architecture where nodes are annotated with a protein, biological pathway, or biological process. We create and employ BINNs on two proteomic datasets, distinguishing between two subphenotypes of septic acute kidney injury (AKI)[29] and COVID-19[30]. The BINNs are trained to classify the subphenotypes based on the proteome as input, whereafter they are interpreted using Shapley Additive Explanations (SHAP)[22], eventually allowing for the identification of important proteins and pathways (Fig. 1). In addition, to demonstrate that the BINNs can be used for different proteomics-platforms, a dataset generated using Olink (Uppsala, Sweden) is analyzed, where a BINN is trained to discriminate between different acute respiratory distress syndrome (ARDS) of different aetiologies[31].

### Construction of biologically informed neural networks

As a starting point, proteomics plasma data from patients suffering from septic AKI and COVID were analyzed to generate datasets for the respective disease. Septic AKI has previously been classified into two subphenotypes of varying severity by latent class analysis of various clinical and molecular markers[32]. In total, 141 samples in the septic AKI training dataset were stratified to one of the two subphenotypes, where 60 samples were classified as subphenotype 1 and 82 as subphenotype 2. Similarly, patients may suffer from varying degrees of COVID-19, which has generated a scale defined by the World Health Organization (WHO) to classify the severity of exhibited symptoms. According to this scale, patients requiring mechanical assistance for ventilation (WHO scale 6–7) are categorized as extremely severe, whereas patients able to breath by themselves as less severe, resulting in two subphenotypes of COVID-19. The COVID training dataset contained a total of 687 samples, where 406 were graded as very severe (WHO scale 6–7) and 281 as less severe (WHO scale < 6). Proteins for both the septic AKI and COVID-19 datasets were quantified using proteotypic peptides from the mass-spectrometry-based proteome maps to ensure unique protein group membership for downstream analysis. The proteomic content of the datasets differed, as 728 proteins were identified in the septic AKI dataset, as compared to the shallower proteome of the COVID-cohort containing 173 proteins.

The datasets were used in combination with the Reactome pathway database[16] to create and train BINNs. As mentioned, the Reactome database contains information about relationships of biological entities, such as molecules, pathways and high-level processes, and does not follow a sequential structure. The underlying graph is therefore subsetted and layerized to fit a sequential neural network-like structure, whereafter it is translated to a sparse neural network architecture, where nodes are annotated with a protein, biological pathway, or biological process - hence biologically informed neural networks. The proteomic content of a sample is passed to the input layer of the network, and the following layers map it to biological processes of increasing level of abstraction—finally ending up in high-level processes such as the immune system, disease, and metabolism. The annotated and sparse nature of the network makes it suitable for introspection and interpretation, as demonstrated by Elmarakeby et al.[25]. The algorithm which uses a graph and a subset of entities to create a sparse sequential neural network was generalized and implemented in the PyTorch framework in Python, and is publicly available at GitHub: https://github.com/InfectionMedicineProteomics/BINN.

Networks for the respective disease were generated with four hidden layers each and differed in architecture due to the discrepancy in the depth of the proteomes of the two datasets—the COVID-BINN being much smaller than the AKI-BINN (Supplementary Fig. 2). Due to their sparse nature, the resulting networks are small—containing trainable parameters in the thousands (AKI-BINN: 6.7 k, COVID-BINN: 1.6 k trainable parameters between hidden layers), as compared to millions which is the case for most contemporary complex deep learning models. The BINNs were trained to identify the subphenotypes of septic AKI and COVID-19 respectively, as outlined above.

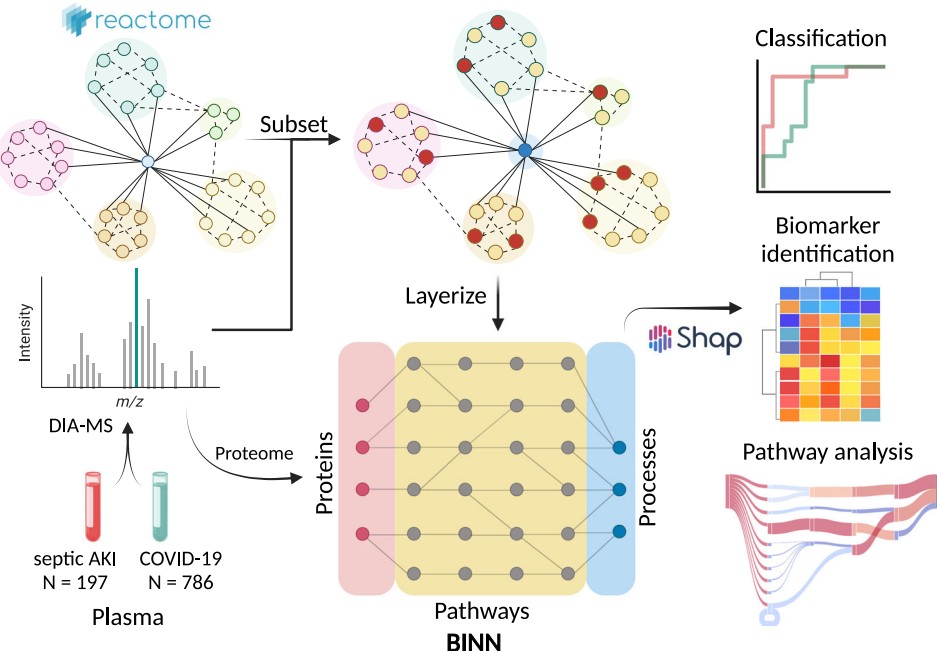

**Fig. 1 | The complete workflow of analyzing proteomic data with biologically informed neural networks.** The plasma proteome from patients suffering from septic AKI and COVID-19 were gathered and analyzed elsewhere[30, 56]. The data was downloaded and re-analyzed, resulting in datasets for the respective disease. The workflow starts by generating a BINN for each dataset by subsetting the pathway database, such as Reactome, using the proteomic content of the dataset of interest and layerizing it to fit a sequential neural network-like structure. The protein quantities for each sample are used to train the respective BINNs to differentiate between two subphenotypes. Thereafter, the networks are interpreted using SHAP and the resulting feature importance values allow for biomarker identification and pathway analysis. Created with BioRender.com.

## Method comparison

To investigate whether machine learning methods were suitable for the stratification of septic AKI and COVID-19-subphenotypes, the BINNs were benchmarked against a support vector machine with radial basis function kernel, k-nearest neighbor, a random forest, and two boosted trees (LightGBM and XGBoost). The evaluation was performed on the complete datasets using k-fold cross-validation ($k = 3$). All machine learning methods achieved AUC scores of $> 0.75$, but the BINNs resulted in the best performances as measured with the area under the receiver operating characteristic curve (ROC-AUC) and under the precision-recall curve (PR-AUC) (ROC-AUC: $0.99 \pm 0.00$ and $0.95 \pm 0.01$, PR-AUC: $0.99 \pm 0.00$ and $0.96 \pm 0.01$) on the septic AKI and COVID-dataset respectively (Fig. 2a–d). Both BINNs achieved a high true positive and true negative rate (septic AKI: $94 \pm 2\%$, $100 \pm 0\%$, COVID: $87 \pm 2\%$, $92 \pm 1\%$) (Fig. 2e, f). The total accuracy for the models were $98.6 \pm 2\%$ (septic AKI) and $87.5 \pm 3\%$ (COVID-19). Additionally, both the AKI and the COVID-BINN attained the highest precision and recall rates out of all methods, achieving a precision of $0.99 \pm 0.020$, $0.87 \pm 0.011$, and recall of $1.0 \pm 0.0$, $0.88 \pm 0.022$, respectively.

To ensure that the measures taken to minimize the risk of overfitting such as the use of dropout, batch normalization and $L2$-regularization, were effective, both the COVID and septic AKI models were tested on independent cohorts. The COVID-BINN was tested on a cohort consisting of 99 samples. These were reported in the same study as the samples comprising the training set, but were gathered at a different hospital[30]. The AKI-BINN was tested on a cohort consisting of 56 samples. These samples were collected in the FINNAKI study[29], but has not been published previously. The COVID-BINN and AKI-BINN achieved accuracies of 87% and 91% respectively on the testing cohorts, confirming that they generalize to unseen data and that overfitting did not occur. This is further motivated by the loss curves, as the evaluated loss during training and validation are matched (Supplementary Fig. 7).

## Interpretation

To identify which proteins, pathways and biological processes were important for the classifications, the trained BINNs were interpreted using SHAP[22]. SHAP is a feature attribution method which estimates the Shapley values (contribution) of each node in the network to the prediction. The node importance can be likened to how much worse predictions were to become after the removal of the said node. SHAP values were adjusted using the logarithm of the number of nodes in the reachable subgraph of a given node to account for the level of connectivity and to remove any biases associated with highly connected nodes (see methods and Supplementary Fig. 5). The node importance of the complete networks were visualized in Sankey diagrams in Fig. 3. Nodes which were given a high SHAP value in the AKI-BINN were largely related to metabolic processes, such as lipid metabolism and those related to PPAR-$a$[33], whereas the COVID-BINN places more importance on nodes related to the immune system and cell death. The emphasis on metabolic processes in the AKI-BINN supports the view of sepsis as a condition with large systemic effects on metabolism, homeostasis and not solely the immune system[34, 35]. In the case of differentiating between COVID severities, processes relating to the immune system (driven by innate immunity), metabolism of proteins, and programmed cell death, seemed to be the most important factors.

**BINN-enhanced biomarker identification.** The first layers of the BINNs contain the proteomic content, and to investigate whether proteins deemed important for the classification by the BINNs could be considered as potential biomarkers, the top-ranking proteins by SHAP value were subject to further investigation. For comparison, a measure of differential expression, the DE score, was devised as a means of standardizing differential expression analysis. The DE score is calculated by scaling the logarithmized fold change and p-value and computing their Pythagorean sum. Proteins which most significantly differ between two groups will therefore be given a high DE score (Eq. (4), Supplementary Fig. 3). Hierarchical clustering using Ward's minimum

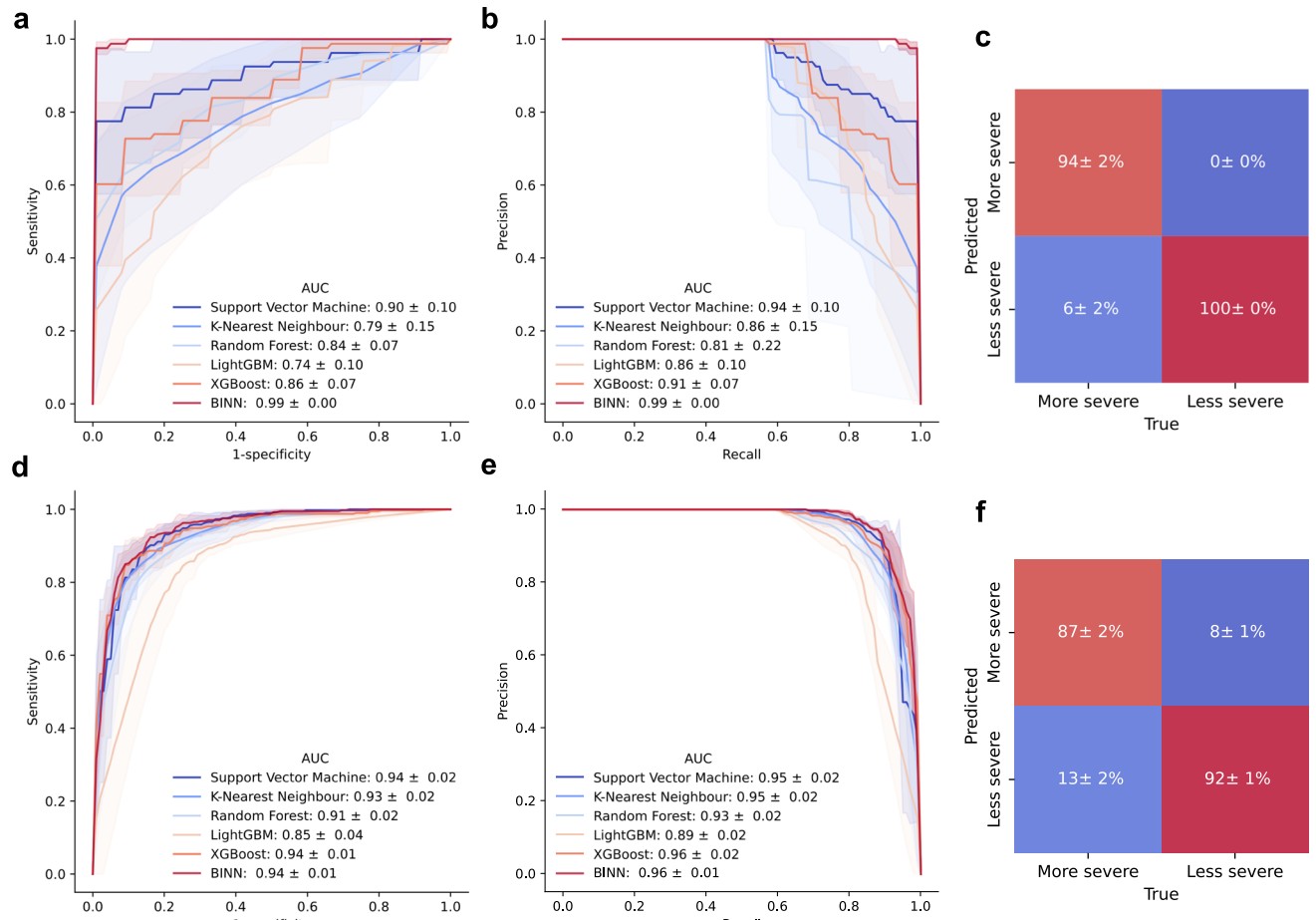

**Fig. 2 | Performance of machine learning methods on the septic AKI and COVID-datasets.** The BINNs and five other machine learning models (support vector machine with radial-basis function kernel, k-nearest neighbor, random forest, LightGBM and XGBoost) were used to predict septic AKI and COVID-19 sub-phenotypes given the proteomic content of the samples. The models were trained and evaluated using k-fold cross validation ($k=3$). **a** The mean ROC-curve and 95% confidence interval for the machine learning methods on the septic AKI dataset. **b** The mean PR-curve and 95% confidence interval for the machine learning methods on the septic AKI dataset. **c** The normalized confusion matrix for the AKI-BINN. The mean ± SD confidence interval is annotated in the matrix. **d** Same as (**a**) but for the COVID dataset. **e** Same as (**b**) but for the COVID dataset. **f** Same as (**c**) but for the COVID dataset. Source data for all panels are provided as a Source Data file.

variance method was performed on the protein quantities of the top 20 proteins identified by SHAP and by DE score in both the AKI and COVID-BINNs.

Several of the top-ranking proteins in the AKI-BINN were known biomarkers for inflammation and have been documented to be altered during severe sepsis, such as CD14[36], FA10[37], H4[38], and OSTP[39]. For example, soluble CD14 has previously been shown to be a promising and rapid responding candidate diagnostic marker for neonatal early and late onset sepsis[40]. Additionally, proteins related to metabolic processes, such as apolipoproteins (APOB, APOA1, APOA2, and APOA4) which also undergo alterations during sepsis[35, 41], were identified. Notably, while the inflammatory markers were included in the top-ranking proteins by DE score, the apolipoproteins were not, and would not be identified with classical differential expression analysis. Clustering on the proteins identified by SHAP resulted in a Rand index of 0.765, outperforming the clustering on proteins ranked by DE score which achieved a Rand index of 0.716 (7.0% increase). Similarly, many of the most important proteins in COVID-BINN have previously been proposed as biomarkers for the distinction between moderately and critically ill COVID patients, such as GELS, ZA2G[42], and S100A8[43]. In the case of COVID, the differential expression analysis resulted in similar proteins and results as the BINN, resulting in Rand indexes of 0.645 and 0.663 respectively when performing hierarchical clustering (2.7% increase).

Markedly, the proteins with the highest SHAP value are not the most significantly differentially expressed or exhibit the highest fold change (see Supplementary Fig. 4). This suggests that some proteins are considered important because of which pathways they are connected to, or due to their co-regulation with other proteins, and would likely have been discarded in typical analyses. Naturally, the proteins selected by DE score differed in relative abundance, although the interpretable machine learning-centered method outperformed differential expression analysis in finding proteins which clustered to the subphenotypes. Clustermaps and plots showcasing the relative abundance of the identified proteins by SHAP can be seen in Fig. 4, and similar plots in the case of differential expression analysis can be seen in Supplementary Fig. 1.

**BINN-enhanced pathway analysis.** Since pathways and processes are integrated into the structure of the BINNs, a subset of pathways may be extracted from the graph underlying the BINN for pathway analysis. One may investigate pathways originating from a certain protein or pathway to see which pathways the node influences, and in turn, which it is influenced by. As mentioned, the candidate diagnostic marker CD14 was identified as one of the most important proteins in the AKI-BINN, and have many known implications in the immune response in general, as well as specifically in the course of sepsis[36, 44]. In Fig. 5a, CD14 has therefore been selected in the AKI-BINN and the downstream

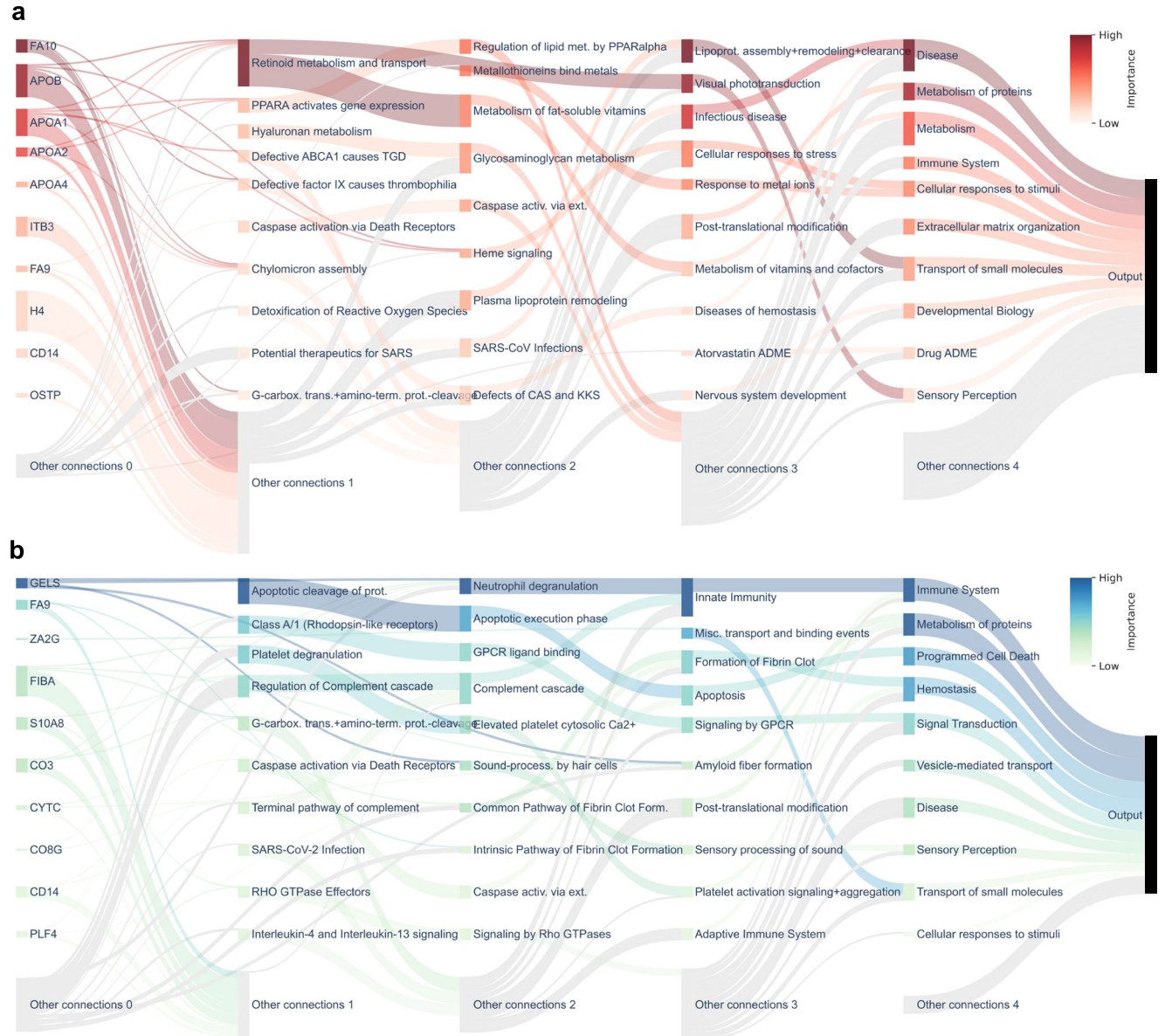

**Fig. 3 | Sankey diagram visualization of node importance in the complete sepsis and COVID-BINNs.** The importance for each node was calculated layer-wise using SHAP and reduced by the level of connectivity, and represented as the outgoing flow from the given node. Node sizes are proportional to the sum of incoming and outgoing values, and therefore take connectivity and importance into account. Additionally, the nodes color reflects its relative importance, as darker nodes are more important in a given layer. The top 10 most important nodes in each layers are showcased and labeled, whereas the rest are gathered in the gray nodes at the bottom of the diagram (labeled "Other connections"). Nodes that had no connection to the labeled nodes i.e., both originated and targeted unlabeled nodes, were discarded for the sake of improved visualization. **a** The AKI-BINN. Nodes related to metabolic processes, such as lipoprotein assembly, remodeling and clearance and metabolism of vitamins and cofactors, and disease, such as infectious disease are considered important in the AKI-BINN. **b** The COVID-BINN. In the COVID-BINN, processes related to immunity, protein metabolism and programmed cell death are dominating. Source data for all panels are provided as a Source Data file.

pathways and processes visualized. In the network, CD14 funnels most of its importance through caspase activation and TLR-associated diseases, and eventually to disease, immune system, and programmed cell death.

Lipoproteins and lipoprotein metabolism are subject to major clinically relevant alterations during sepsis[35, 41, 45], and indeed many lipoproteins and related pathways and processes were identified in the AKI-BINN, as described above. When inspecting the subgraph upstream from plasma lipoprotein remodeling, LDL remodeling and APOB, APOA1, APOA2 and APOA4 can be identified as the most important sub-process and proteins respectively (Fig. 5b).

GELS has previously been identified to play an important role in various physiological conditions, diseases, and inflammatory processes[46], and was identified as one of the most important proteins

in the COVID-BINN. After inspection of the subgraph originating from GELS, we identify that it contributes mostly to *apoptotic cleavage of* proteins and neutrophil degranulation–processes which eventually contribute to programmed cell death and the immune system (Fig. 5c). Both neutrophil degranulation[47] and programmed cell death[48] have been found to be pivotal in the course of severe COVID-19.

Pathway analysis plays a key role in understanding complex biological systems, and is naturally closely tied to proteomic content. To compare the integrated pathway analysis utilizing BINNs with common contemporary methods, pathway analysis with Metascape was performed[15]. This resulted in largely the same set of pathways ranking highly in both the COVID-19 and sepsis datasets, a majority of which are related to the inflammatory response (Supplementary Fig. 6). Utilizing the interpretable nature of the BINNs and querying their

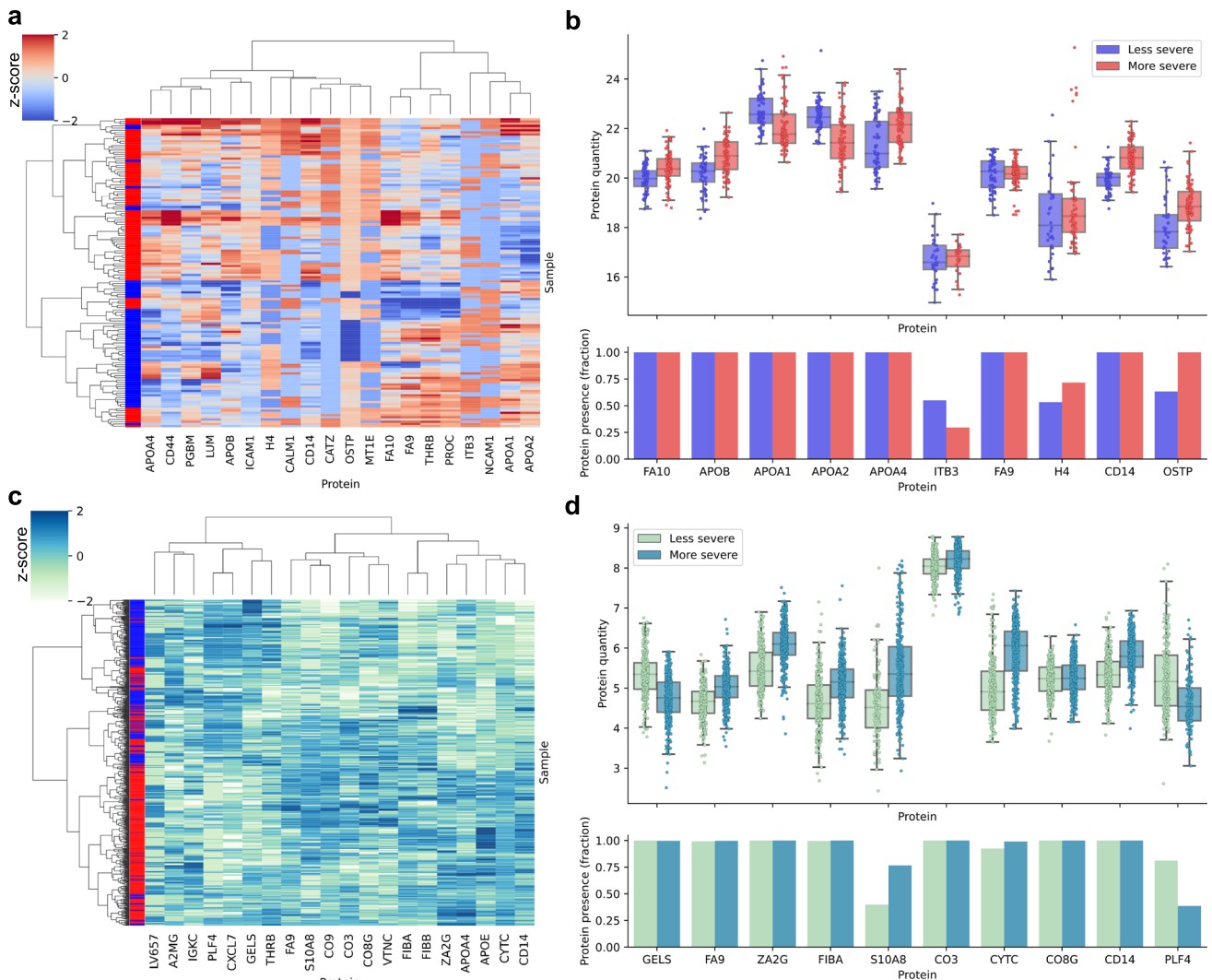

**Fig. 4 | Clustering on the proteins with the highest SHAP values in the septic AKI and COVID-datasets.** The most important proteins as determined by the BINNs were selected and subject to hierarchical clustering. **a** A clustermap showcasing the clustering based on the scaled protein abundances of the top 20 most important proteins in the AKI-BINN. The left-most column shows the subphenotype classification (subphenotype 2: red, subphenotype 1: blue). Clustering was performed using Wards minimum variance method and Euclidean distances. The Rand index for the clustering was 0.765. **b** The upper panel shows the protein quantity for the 10 most important proteins. The boxes show the quartiles of the distribution while the whiskers extend to show the rest of the distribution, except for points that are determined to be outliers using a method that is a function of the inter-quartile range. The center-line shows the mean of the dataset. $n = 141$ biologically independent samples. The lower panel shows in which fraction of the samples the given protein was identified. When training the predictors, proteins which were not identified in a sample were imputed with a 0. **c** Same as (**a**) but on the COVID dataset. The Rand index for the clustering was 0.663. **d** Same as (**b**) but for the COVID dataset. Here $n = 687$ biologically independent samples. Source data for all panels are provided as a Source Data file.

underlying graphs allowed us to find important pathways and relationships that were omitted when using contemporary methods, highlighting the advantages of the BINNs for custom pathway analysis.

### Cross-platform generalizability

To demonstrate the ability of the BINN to generalize cross-platform, a proteomics dataset generated using the Olink-platform (Uppsala, Sweden) was analyzed[31]. Here, the proteomic content of urine from patients suffering from bacterial sepsis-induced ARDS (17 samples) and COVID-19-induced ARDS (42 samples) were analyzed. A cohort of healthy controls was also included (25 samples). The pre-processed data data was downloaded and used without modifications.

A BINN was generated in the same manner as for the AKI and COVID-BINNs. The Olink BINN was trained to classify between the three classes: COVID-19-induced ARDS, bacterial sepsis-induced ARDS and healthy controls, and was evaluated using $k$-fold cross validation ($k = 3$). This is a three-class classification problem with a low number of

samples, however, the Olink-BINN still managed to identify healthy and COVID-19-induced ARDS with high accuracy (true positive rates: healthy: $0.8 \pm 0.12$, COVID-19-ARDS: $0.81 \pm 0.03$). The low number of samples and the heterogeneity of the bacterial sepsis-induced ARDS resulted in a low true positive rate for this class ($0.29 \pm 0.15$) (Fig. 6). Additionally, the BINN highlighted several pathways with relevance to ARDS such as the G-coupled protein receptors-pathway[49], which contributed to the Signal Transduction pathway as being one of the most important (Fig. 6).

### Discussion

We present and apply a generalized workflow utilizing biologically informed neural networks (BINNs) and feature attribution methods for biomarker discovery and pathway analysis from different types of proteomics data. Although the BINNs are sparse and have few trainable parameters, they accurately predicted degrees of severity in both septic AKI and COVID-19 from the plasma proteome alone. The sparse

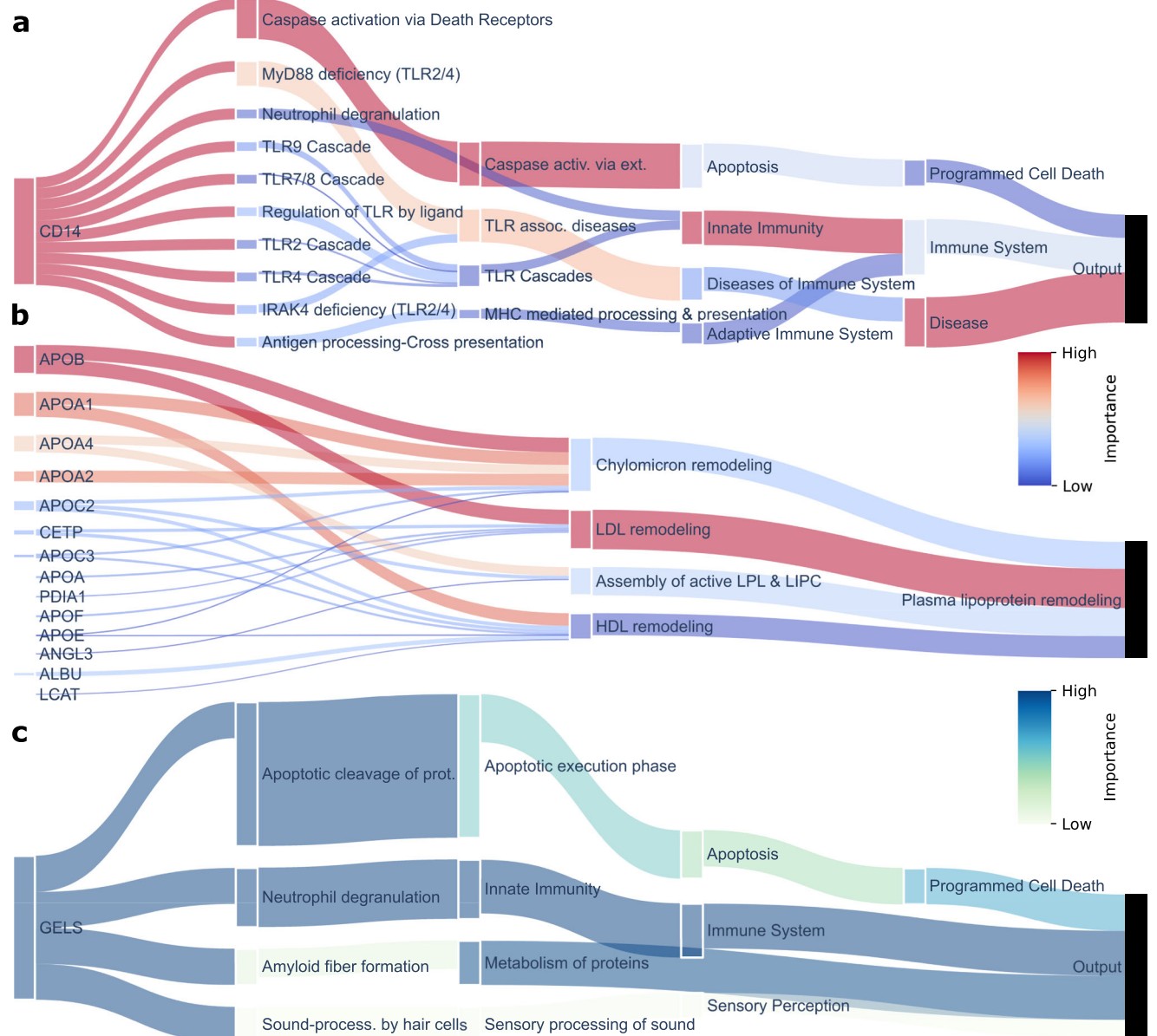

**Fig. 5 | Custom pathway-analysis utilizing the interpreted BINNs.** The graph underlying the interpreted BINNs can be extracted and subsetted for custom pathway analysis. **a** The down-stream graph originating from CD14 in the AKI-BINN. The most important contribution of CD14 is to *caspase activation via death receptors,* MyD88 deficiency, and subsequently, disease and programmed cell death. **b** The up-stream graph originating from plasma lipoprotein remodeling. Its most important contributor is LDL remodeling, HDL remodeling and four apolipoproteins: APOB, APOA1, APOA4, and APOA2. **c** The down-stream graph originating from GELS in the COVID-BINN. GELS eventually connects to programmed cell death, sensory perception, immune system, and metabolism of proteins where programmed cell death and immune system are the most important high-level processes and sensory perception has little impact on the network. Source data for all panels are provided as a Source Data file.

and informed nature of the BINN incorporates biological pathways and processes into its architecture, tailoring it for introspection. Further, biological relationships which are typically overlooked in common methods are captured in the network, and therefore highly relevant information is incorporated into the analysis. Ultimately, this allows for a comprehensive analysis of proteomic data in a single unified method.

Interpreting the BINNs trained to predict different subphenotypes of septic AKI and COVID-19 identified several relevant biomarkers and pathways that were omitted when using common methods of differential expression and pathway analysis. Furthermore, it highlighted key differences between the two diseases, as proteins and processes related to *metabolism* and *disease* were considered highly important in the AKI-BINN, whereas the COVID-BINN favored proteins and processes related to *immunity*.

Biomarker discovery in the context of BINNs is performed by calculating the feature importance of the initial layer of the network. Several of the most important proteins in the sepsis and COVID-BINNs were known biomarkers of the respective disease, however, they differed from the most differentially expressed proteins. Important proteins were not necessarily the most significantly differentially expressed (Supplementary Fig. 4). Proteins highlighted in the AKI-BINN were both prognostic inflammatory biomarkers such as CD14, FA10, and OSTP, but also biomarkers related to metabolic proteins such as apolipoproteins. Some apolipoproteins have been found to correlate to 30-day mortality in sepsis, as well as platelet activation and monocyte activation affecting patient outcomes[45] and were not among the top proteins by differential expression. The COVID-BINN highlighted several proteins which have been proposed as diagnostic

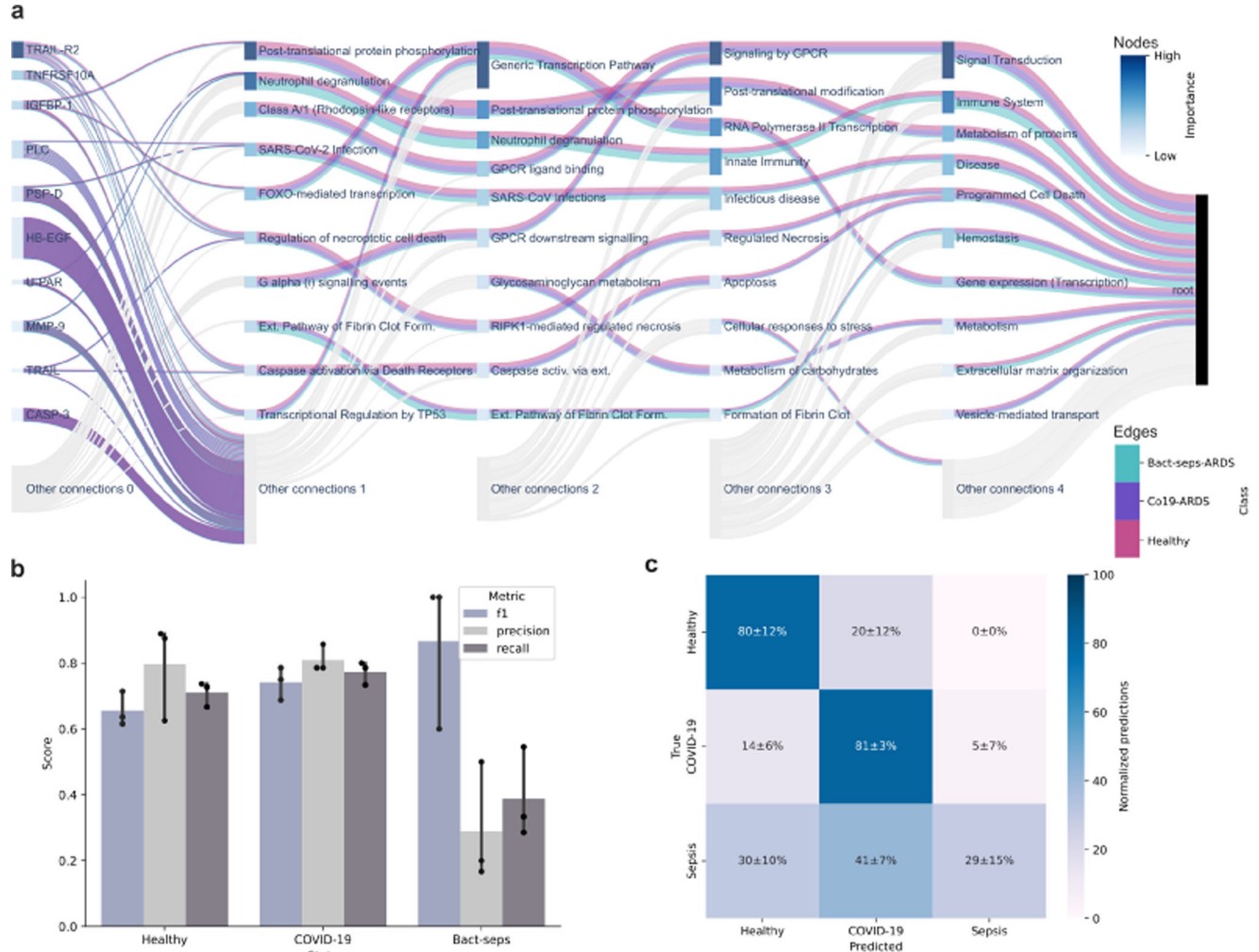

**Fig. 6 | A BINN trained and constructed from Olink-data.** To demonstrate the ability of the BINN package to generalize cross-platform, a BINN was constructed using an proteomic dataset generated using the Olink platform and the Reactome pathway database[31]. The Olink-BINN was trained to differentiate between COVID-19-induced ARDS, bacterial sepsis-induced ARDS and healthy controls. **a** The resulting BINN. Since this is a three-class classification, the connections are colored based on which class the SHAP value pertains to. The flow is partitioned into the three classes, allowing us to identify which nodes are important for classifying a particular class. For example, *Neutrophil degranulation* is important during the classification of bacterial sepsis-induced ARDS and healthy controls, whereas Post translatuional protein phosphorylation is mostly important for COVID-19-induced ARDS. The node colors still reflect the mean importance and the nodes are ordered accordingly. **b** The average f1-score, precision and recall of the Olink-BINN for the different classes during validation with k-fold cross validation ($k = 3$). Error bars show the 95% confidence interval. **c** Confusion matrix averaged across folds and normalized per true label (rows sum to 100%). The mean and 95% confidence interval is annotated in the matrix. Source data for all panels are provided as a Source Data file.

markers for critically ill COVID patients, such as GELS, ZA2G, and S100A8[42,46,50]. Notably, levels of GELS and S100A8 in plasma and have been found to be directly correlated to the severity of COVID-19[43,50]. Biomarker identification with BINNs and classical differential expression may be seen as complementary approaches, and both may provide value to an analysis. Whereas differential expression analysis is guaranteed to provide proteins with a high fold change and low *p*-value, as this is the selection criteria, a BINN will provide the proteins which are important in a classification context when taking biological processes into account.

The major strength of BINNs lies in their embedding of pathway analysis into the architecture as the graph underlying the trained network can be extracted and subsetted to identify influential nodes in the subgraphs. This enables the investigation of downstream pathways from a given protein to understand the extent of its impact in the network. Similarly, the proteins and pathways upstream from a given node can be extracted to identify the extent of their influence. Comparatively, this provides a major

improvement to how generic pathway analysis is commonly performed in proteomic research, where proteins associated with pathways are counted and the pathway with the most connections is considered the most relevant.

The performances of the BINNs relative to other machine learning methods differed between the datasets, as the performance of the COVID-BINN was comparable to other methods, while the AKI-BINN outperformed other methods (Fig. 2). This is likely due to the combination of a higher dimensionality and smaller cohort-size of the sepsis dataset, suggesting that the BINNs are able to represent the feature space more accurately in complex datasets given fewer examples as compared to shallower learning-methods. Beyond performance, the varying proteome depths may also have implications on the conclusions drawn after interpreting the networks, as the underlying proteomes influences their architectures. Such effects should be kept in mind when comparing networks, as was done when identifying metabolic processes to be more important in the AKI-BINN than the COVID-BINN.

The ability of a model to generalize to new data depends on the quality, diversity, and size of training datasets to capture the underlying distributions of the data. Adequate dataset size can help prevent overfitting, and provide coverage of various scenarios to facilitate real-world applicability. To evaluate how BINNs generalize to new data, we provide previously unseen test sets for both the AKI and COVID-BINNs. The high accuracies of both the AKI and COVID-BINNs (91% and 87%, respectively) suggest their ability to generalize to unseen samples effectively. However, since the number of samples in biological experiments are typically decided based on availability, the training sets used in our study may not fully represent the complete underlying distributions, and could be expanded to maximize the potential of BINNs based on the experiment.

It was found that hyperparameter configuration had a significant influence on the distribution of importance in the network. Specifically, prolonged training durations resulted in a dependency on combinations of low abundance features such as antibodies, which although improved classification accuracy, are of less biological interest in this context. The BINN is highly dependent on the quality of the underlying graph, the dataset as well as the overlap with the dataset. Proteins which are not mapped to events in the Reactome pathway database are discarded in the analysis, and for small datasets the reduction in features may be detrimental. Unsupervised learning methods aimed at classifying nodes such as BIONIC[51] may be utilized to generate comprehensive networks encompassing a large majority of the proteome which could be used to generate BINNs. However, defining and annotating processes and pathways is still a manual and laborious task limiting the size of the BINNs. Our implementation is agnostic to the underlying graph and inputs used for the creation of the network, allowing for e.g., genomic or metabolomic data to be used in combination with different pathway repositories such as KEGG[52], GeneOntology[53, 54], or a custom curated set of pathways, to generate BINNs. In addition, the BINN package can be used to analyze data from different platforms. The ability for BINNs to analyze data from different proteomics platforms and with different types of underlying graphs extends its reach in biomedical and clinical applications including biomarker discovery, drug target discovery and subphenotype classifications as these problems are highly multifaceted.

In summary, we demonstrate how BINNs can be trained, interpreted and visualized to provide a comprehensive analysis of proteomic datasets. The methodology behind the creation, analysis, and visualization of interpreted BINNs has been generalized and is publicly available, opening up possibilities for further analyses and development in the realm of machine learning and proteomics.

## Methods

### Data

Blood plasma from patients suffering from septic AKI and COVID-19 were gathered and analyzed elsewhere, whereafter the resulting proteomic datasets were uploaded to proteomeXchange[55] and made publicly available[30, 56]. The COVID-19 dataset consisting of the raw data matrix of quantified precursors and design matrix with patient annotations were downloaded from PRIDE (PXD025752)[57] and re-analyzed. The original study reports two cohorts from different hospitals whereof the samples gathered at Charité containing 687 samples were used for training, and the samples gathered at Innsbruck consisting of 99 samples were used as a testing cohort[30]. These are available under the same PRIDE identifier. The raw mass spectrometry files and spectral library for the septic AKI training dataset with 141 samples were downloaded from PRIDE (PXD038394) and analyzed with an adapted version of the DIAnRT workflow[58] using GPS[56] for validation. Using OpenSwath (v. 2.6)[59], a first iteration of sub-optimal retention time alignment is performed followed by validation and refined retention time alignment using the highest scoring quantified precursors for each run. This process is repeated 3 times, with strict retention time alignment and mass correction on the final iteration followed by false discovery rate control at the global peptide and protein levels to generate a quantitative matrix.

To generate a testing dataset for the sepsis model, 56 previously unpublished samples from the FINNAKI[29] study were processed. The protocol for sample preparation and data acquisition is identical to how the previously published septic AKI dataset was generated by Scott et al.[56]. All sample preparation steps of the 56 samples, including desalting and protein digestion, used the Agilent AssayMAP Bravo Platform (Agilent Technologies, Inc.) per manufacturer's protocol. Each plasma sample was diluted 1:10 (100-mM ammonium bicarbonate (AmBic); Sigma-Aldrich Co, St Louis, MO, USA), and 10 l of each diluted plasma sample were transferred to a 96-well plate (Greiner G650201) where 40 $\mu$L of 4 M urea (Sigma-Aldrich) in 100 mM AmBic was manually added with a pipette for a final volume of 50 $\mu$L. The proteins were reduced with 10 $\mu$L of 60 mM dithiothreitol (DTT, final concentration of 10 mM, Sigma-Aldrich) for one hour at 37 °C followed by alkylation with 20 $\mu$L of 80 mM iodoacetamide (IAA, final concentration of 20 mM, Sigma-Aldrich) for 30 min in a dark at room temperature. The plasma samples were digested with 2 $\mu$g Lys-C (FUJIFILM Wako Chemicals U.S.A. Corporation) for five hours at room temperature and further digested with 2 $\mu$g trypsin (Sequencing Grade Modified, Promega, Madison, WI, USA) overnight at room temperature[60]. The digestion was stopped by pipetting 20 $\mu$L of 10% trifluoroacetic acid (TFA, Sigma-Aldrich) and the digested peptides were desalted on Bravo platform. To prime and equilibrate the AssayMAP C18 cartridges (Agilent, PN: 5190-6532), 90% acetonitrile (ACN, Sigma-Aldrich) with 0.1% TFA and 0.1% TFA were used, respectively. The samples were loaded into the cartridges at the flow rate of 5 $\mu$L/min. The cartridges were washed with 0.1% TFA before the peptides were eluted with 80% ACN/0.1% TFA. The eluted peptides were dried in a SpeedVac (Concentrator plus Eppendorf) and resuspended in 25 $\mu$L of 2% ACN/0.1% TFA. The peptide concentration was measured using the Pierce Quantitative Colorimetric Peptide Assay (Thermo Fisher Scientific, Rockford, IL, USA). The samples, 10 $\mu$L, were diluted with 10 $\mu$L 2% ACN/0.1% TFA and spiked with synthetic iRT peptides (JPT Peptide Technologies, GmbH, Berlin, Germany) before liquid chromatography-mass spectrometry (LC-MS/MS) analysis.

The AKI samples were processed in accordance with the Helsinki Declaration. The Ethics Committee of the Department of Surgery, Helsinki and Uusimaa Hospital District, approved the study protocol and each participant or their proxy gave written informed consent. The Ethics Committee of the Department of Surgery, Helsinki and Uusimaa Hospital District, also approved the inclusion of participants for all centers involved as well as the use of deferred consent (Reference Number 18/13/03/02/2010). Patient demographics, medical history, severity scores, length of stay, physiologic data and hospital mortality were collected from the Finnish Intensive Care Consortium prospective database (Tieto Ltd, Helsinki, Finland) with a study-specific case report form.

Peptide analyses were performed on a Q Exactive HF-X mass spectrometer (Thermo Fisher Scientific) connected to an EASY-nLC 1200 ultra-HPLC system (Thermo Fisher Scientific). Peptides were trapped on precolumn (PepMap100 C18 3 $\mu$l; 75 $\mu$l × 2 cm; Thermo Fisher Scientific) and separated on an EASY-Spray column (ES903, column temperature 45 °C; Thermo Fisher Scientific). Equilibrations of columns and sample loading were performed per manufacturer's guidelines. Mobile phases of solvent A (0.1% formic acid), and solvent B (0.1% formic acid, 80% acetonitrile) was used to run a linear gradient from 5% to 38% over various gradient length times at a flow rate of 350 nl/min. The 44 variable windows DIA acquisition method is described by Bruderer et al[61]. MS raw data was stored and managed by openBIS (20.10.0)[62] and converted to centrioded indexed mzML files with ThermoRawFileParser (1.3.1)[63].

To demonstrate the ability of the BINN to generalize cross-platform, a third proteomics dataset generated using the Olink platform

(Uppsala, Sweden) was analyzed[31]. The pre-processed data were downloaded from an online repository supplied in the original study[31] and analyzed without modifications. The dataset contains 84 samples generated by analyzing the urine of patients suffering from sepsis-induced ARDS, COVID-19-induced ARDS, as well as healthy controls. This dataset is henceforth referred to as the Olink dataset.

## Data processing

The septic AKI and COVID-19 DIA datasets were processed in the same manner using the open source python package DPKS (https://github.com/InfectionMedicineProteomics/DPKS). The quantitative matrices were filtered to remove decoys and precursors that did not pass a 1% false discovery rate control at the global peptide and protein levels. Samples were then mean-normalized to remove any bias in the data and proteins were quantified from proteotypic peptides using a python implementation of the relative quantification *iq*-algorithm[64]. Differential expression analysis was performed between each group of each dataset for proteins quantified in a minimum of 3 samples per group using linear models and multiple testing correction with DPKS. For input into the BINN, only proteins considered in the differential analyses were used as input, and missing values were imputed as 0.

## BINN

The BINN was first introduced as P-NET by Elmarakeby et al.[25], and the architecture and methodology closely resemble the one they presented. Here, however, we introduce a generalized methodology as demonstrated in the context of proteomics analysis and present further applications of the informed network. The BINN is a sequential sparse feed-forward neural network which is generated using an underlying graph. The underlying graph used in this study is that of the Reactome pathway database[16] and contains information about relationships of biological entities, such as molecules, pathways and high-level processes. The graph is processed and layerized before it is translated into neural network in the PyTorch framework[65]. The generalized algorithm underlying the creation of a BINN from the Reactome pathway database was implemented as a Python package:

1. Subset the Reactome pathways database (directed graph) using the union of proteins by adding the parental pathway, starting at the protein level, until the highest level of nodes is reached (nodes with out degree = 0).
2. Generate a network from the subsetted pathways and add an output node connected to the highest level of nodes. The number of output nodes correspond to the number of classes the network is set to predict.
3. Starting at the output node, traverse the network backwards for *N* layers If reaching a terminal node before *N* layers have been reached−add a copy of the previous node. This implies that the path depth $\leq N + 1$.
4. Remove nodes which have not been traversed.
5. Connect proteins to the final corresponding terminal nodes.

The constraints on connectivity substantially decrease the number of trainable parameters in a BINN, resulting in smaller networks than contemporary architectures. In this study, three networks were generated, originating from three different proteomics datasets: the first being analyzed blood plasma from patients suffering from septic AKI, the second from patients suffering from COVID-19 and the third from patients suffering from either sepsis-induced or COVID-19-induced ARDS. The sepsis and COVID-19 datasets contained a total of 1203 and 174 proteins respectively, while the Olink dataset contained 265 proteins. All proteins were not present in the minimum requirement of 3 samples per group or were not present in the Reactome database, reducing the final number of proteins to 728 (septic AKI), 127 (COVID-19) and 230 (Olink). The Reactome pathway database was downloaded 2022-07-14. When generating networks with 4 layers,

this resulted in 6.7 thousand (septic AKI), 1.6 thousand (COVID-19) and 1.4 thousand (Olink) trainable parameters between the hidden layers in total (Supplementary Fig. 2).

The network is constructed so that the lowest level of entities exists in the input layer, and the level of abstraction increases as the network is traversed towards the output layer. The first layer (input layer) therefore contains the proteins, and are fed the scaled protein abundances. Thereafter follows the lower-level biological pathways from the Reactome database, such as regulation of the complement cascade. The final layer contains information about high-level biological processes, such as immune system, hemostasis, disease, and metabolism. The hidden linear layers are intercepted by *tanh*-activation layers, as well as dropout layers and batch normalization.

The BINN is interpreted using SHAP[22]. SHAP is a feature attribution method which computes the importance of a given feature to the outcome of the model. Similar to LIME[24], SHAP applies a linear relationship in its explanation model. Furthermore, the properties of the feature importance values are equivalent to the properties of the well-established Shapley values[66], which, in short, makes SHAP a feature attribution method which estimates Shapley values with a linear explanation model. SHAP provides a range of kernels which can be used for various models, one of which being the Deep SHAP kernel, which similar to DeepLift[23] can be applied to deep learning models such as neural networks. In essence, Deep SHAP improves on the DeepLift algorithm, by approximating the conditional expectations using a set of background samples. Thereafter, the SHAP values can be approximated such that they sum to the difference of the expected model output (based on the set of background samples) and the current model output: $f(x) - E(f(x))$.

Problems arise if one wants the node importance to be meaningful for all layers in a sequential feed-forward neural network. This is because earlier nodes may completely rely on later nodes, and may not be important by themselves. For the node importance to reflect that which is both important in itself, and important in the context of the complete network, fully connected output layers are placed after each hidden layer, and the final prediction is computed as the average of all of the output layers. The output from each output layer is passed through a $\sigma$-activation function before being averaged.

$$\text{out}_{\text{final}} = \frac{\sum_{\text{layer}=0}^{N} \sigma(\text{out}_{\text{layer}})}{N} \tag{1}$$

Nodes that are highly connected may be given an importance score which does not reflect its biological importance, but is an artifact of the architecture. Elmarakeby et al.[25] used the graph informed function, *f*, to reduce bias that may be introduced by over-annotation of certain nodes:

$$d_{\text{tot}_n} = d_{\text{in}_n} + d_{\text{out}_n}$$

$$f(S_n) = \begin{cases} \frac{S_n}{d_{\text{tot}_n}}, & d_{\text{tot}_n} > \mu + 5\sigma \\ S_n, & \text{otherwise} \end{cases} . \tag{2}$$

here $d_{\text{in}_n}$ and $d_{\text{out}_n}$ are the in degree and out degree of a given node, *n*. To motivate the use of a bias reduction technique like this, we'd expect to see a correlation between the node degree and importance value. We suggest that a more general measure of node influence is the number of nodes in the complete subgraph defined by node *n*, $N_{SG_n}$. The complete subgraph of node *n* is defined as the complete set of predecessors and successors originating from *n* in the directed graph *G*. The outgoing and incoming edges may be seen as a proxy for the size of $SG_n$. The connections in a fully connected graph grows exponentially with the number of nodes, and

**Table 1 | An overview of the data used in this study**

| Model | # samples | Origin | Usage | Available at |
|---|---|---|---|---|
| sepsis-BINN | 141 | FINNAKI study, previously published[29] | Training/validation | PRIDE (PXD038394) |
| sepsis-BINN | 56 | FINNAKI study, published in our study | Testing | PRIDE (PXD044264) |
| covid-BINN | 687 | Charité hospital[30] | Training/validation | PRIDE (PXD025752) |
| covid-BINN | 99 | Innsbrück hospital[30] | Testing | PRIDE (PXD025752) |
| olink-BINN | 84 | Weill Cornell Biobank of Critical Illness | Training/validation | Available in original study[31] |

$\log(N_{SG_n})$ may therefore be an appropriate reduction factor. Calculating the Pearson correlation coefficient for the mentioned graph informed measures and the SHAP value shows that the $N_{SG_n}$ and $\log(N_{SG_n})$ indeed are the graph informed functions that are most correlated with SHAP value, although this varies between layers (Supplementary Fig. 5). The adjusted node importances may therefore be calculated by:

$$f(S_n) = \frac{S_n}{\log(N_{SG_n})} \tag{3}$$

### Training and evaluation

The generated datasets were scaled so that the distribution had a mean of 0 and variance of 1. The two BINNs generated using the DIA datasets were trained and evaluated on the respective dataset using k-fold cross-validation ($k = 3$) alongside five machine learning models (support vector machine with radial-basis function, K-nearest neighbor, random forest, LightGBM and XGBoost). Their performances were evaluated using the area under the receiver operating characteristic (ROC) curve and the area under the precision-recall (PR) curve. The area under the curves were averaged across cross-validation runs. The BINNs were trained until halted using early stopping and with similar hyperparemeter configurations. The learning rate was initiated at 0.001, and decreased adaptively if the validation loss plateaued. The networks seek to minimize the cross-entropy error with an Adam-optimizer. A weight-decay ($L2$-penalty) of 0.001 is applied during training. Several measures were taken to mitigate the risk of overfitting, such as the use of drop-out, as well as the adaptive learning rate, training times, and penalties mentioned above. Additionally, both BINNs were tested on independent testing cohorts which were never seen during training to ensure that the measures taken to reduce the risk of overfitting were effective.

When generating models for interpretation, the BINNs were trained on the complete dataset, and never validated. In such cases, the adaptive learning rate is monitoring the training loss instead of the validation loss. Training was halted when training loss plateaued. The evaluation time is dependent on the number of background samples used and it is often necessary to use a subset of the dataset as background, however, due to the relatively small number of samples in the dataset, the complete datasets were used to evaluate $E(f(x))$.

The dataset generated using the Olink platform was used to create an Olink-BINN. The Olink BINN had the same hyperparameters as defined above, except for an increased initial learning rate of 0.01. It was validated using k-fold-cross validation ($k = 3$) and interpreted as per above.

### Biomarker evaluation and pathway analysis

Proteins, pathways, and processes deemed important for the network during classification will be the ones that contribute greatly to correct predictions. The interpreted network can therefore be introspected and used for biomarker evaluation and pathway analysis. The proteins deemed important in the first layer can be extracted and compared to the ones that are differentially expressed. To get a quantitative measure of differential expression for a protein, $p$, the following expression was devised:

$$DE_p = \sqrt{\frac{FC_p}{\max(FC)}^2 + \frac{\log(p\text{-value}_p)}{\min(\log(p\text{-value}))}^2} \tag{4}$$

This normalizes the fold change (FC) and $\log(p$-value) and calculates the Euclidean distance from *origo* (i.e., the Pythagorean sum). One can visualize this measure as the distance from *origo* in a volcano plot with a standardized scale on the $x$ and $y$-axis (Supplementary Fig. 3). The 20 proteins with the highest SHAP value and the highest DE-value were subject to hierarchical clustering using Ward's minimum variance method.

One can subset the graph underlying the BINNs to extract subgraphs originating from, or incoming to, a certain node. The interpreted subgraph can be used for pathway analysis to gain further understanding of the dataset. We implemented three ways to subset the graph: downstream, upstream, and the combined downstream and upstream (complete subgraph). In a downstream subgraph, the pathways originating from a certain node is included, whereas in the case for an upstream subgraph, the nodes eventually reaching a certain node are included. A complete subgraph can be seen as the union of both the downstream and upstream subgraph.

### Statistics and reproducibility

When evaluating the performance of BINNs and other machine learning methods k-fold cross-validation was conducted ($k = 3$) and area under the receiver operating characteristic curve as well as under the precision-recall curve is presented as the mean ± standard deviation. Additionally, two testing datasets are included and applied to evaluate the possibility of the BINNs overfitting. In testing scenarios, the BINNs are trained on the complete training dataset and evaluated on the test set. A total of five datasets are used in this study, four of which are previously published (see Table 1). For all datasets, all samples available in the dataset were used here, and no data were excluded from the analyses. In the novel septic-AKI test set published here, the dataset size was determined based on availability of biological samples, and no statistical method was used to predetermine sample size. Beyond the randomization during k-fold cross validation, the experiments were not randomized. When calculating change in differential expression, the p-values are calculated using linear least-squares regression and multiple testing correction is done using 2-stage false discovery rate Benjamini–Hochberg method[67] as implemented in the processing package DPKS https://github.com/InfectionMedicineProteomics/DPKS. SHAP values were calculated for each layer in the BINNs using the Deep SHAP algorithm[22]. In all cases, the complete datasets were used as background data to establish the average output of the networks.

### Reporting summary

Further information on research design is available in the Nature Portfolio Reporting Summary linked to this article.

## Data availability

All relevant data supporting the key findings of this study are either downloaded from open repositories or have been uploaded to such

repositories and are publicly available. The previously published DIA-MS septic AKI dataset is available with the PRIDE accession code PXD038394 [https://www.ebi.ac.uk/pride/archive/projects/PXD038394]. The COVID-19 dataset is available with the PRIDE accession code PXD025752 [https://www.ebi.ac.uk/pride/archive/projects/PXD025752]. The Olink dataset is available from: https://doi.org/10.6084/m9.figshare.20260998.v1. The previously unpublished DIA-MS septic AKI dataset has been deposited to the ProteomeXchange Consortium via the PRIDE partner repository with the identifier PXD044264 [https://www.ebi.ac.uk/pride/archive/projects/PXD044264]. The Reactome Pathway Database was downloaded from: https://reactome.org/download-data in July 2022. Source data are provided with this paper.

## Code availability

The complete code behind the BINN-package is available as a GitHub repository under an MIT license: https://github.com/InfectionMedicineProteomics/BINN. The package has been deposited at Zenodo (https://zenodo.org/record/8207421)[28] with documentation at https://infectionmedicineproteomics.github.io/BINN/. The DE-score algorithm has been implemented into the data processing package DPKS which is available under an MIT license at: https://github.com/InfectionMedicineProteomics/DPKS.

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

## Acknowledgements

J.M. was supported by the Wallenberg foundation (WAF grant number 2017.0271), the Swedish research council (grant number 2019-01646 and 2018-05795) and Alfred Österlunds Foundation. A.L. was supported by ALF project funding.

## Author contributions

E.H. and A.S. both contributed to the ideation and implementation of the work, as well as the writing of the manuscript. J.M. and L.M. contributed conceptualization and insights to the project, and contributed to writing of the manuscript. C.K., T.M., A.L., and S.T.V. all contributed to gathering and processing the previously unpublished septic AKI samples.

## Funding

## Competing interests

The authors declare no competing interests.
