## [Peer Review File · Nature Communications]

Interpreting biologically informed neural networks for enhanced proteomic biomarker discovery and pathway analysisReviewer #1 (Remarks to the Author):

In this manuscript, the authors implemented an explainable artificial intelligence (XAI) framework of biologically informed neural networks (BINN), by mirroring the complex molecular relationships curated in the Reactome pathway database as the connections of neurons. Using this BINN, they compiled two plasma proteomic data sets for two immune-related diseases, sepsis induced acute kidney injury (AKI) and COVID-19 disease. Using the n-fold cross-validation, the classification accuracy of BINN was compared to other existing method, with much higher AUC and AUPR values. Two major flaws existed in the current form, which must be carefully addressed prior to further consideration.

1. The technical aspect. The authors claimed that AUC and AUPR values of BINN reached to 0.99 and 0.99 for septic-AKI, and 0.95 and 0.96 for COVID-19 disease. The authors must exclude the possibility of over-fitting. Unlike genomic or transcriptomic profiling, the fluctuation of proteomic quantification is highly dynamic. For example, if a patient just drinks a small cup of waters, the plasma proteomic quantification might undergo a considerable change. And due to a biological process, circadian rhythm, the proteomic compositions will be significantly fluctuated in morning and night for a single patient. Also, the coverage of proteomic profiling is quite low, and many proteins detectable in some experiments will be missed in other runs. So imputation is needed to complement the data, and different imputation methods influence the identification of protein biomarkers. Third, the genetic background of people worldwide is not identical, and proteomic compositions are obviously different in people. So for a highly dynamic and unintegrated data type, how a machine learning approach can reach such a high accuracy? The only explanation is that the computational model is highly over-fitted. To evaluate the robustness of BINN, the n-fold cross-validation is not enough, and the authors must collect published cohorts published by other groups. Such published data in either sepsis or COVID-19 are very easy to be obtained, and these data should be used as independent data set not used for training. In addition, the authors should collect an enough number of proteomic data from healthy volunteers as the control, not only just classification of mild or severe disease.

2. The biological aspect. A predictor with a nearly 100% accuracy means that all factors, including genetic differences, life styles, and geographical features, as well as real proteomic changes, were almost fully captured. The authors should explain what they observed from SHAP interpretation, beyond the proteomic signatures. Also, due to the much higher performance of BINN, it can be expected that new biomarkers can be identified from the sample classification. Without additional experimental validations is OK, however, the authors should put their findings in the context of the literature, to demonstrate why the newly predicted biomarkers are truly involved in septic-AKI or COVID-19.

Taken together, my overall thinkings are: 1) The BINN framework is interesting and potentially used for sample classification and biomarker finding, in a higher interpretable manner; 2) Additional cohorts not used for training should be compiled from published literature for a justified testing and comparison. 3) Newly uncovered protein biomarkers should be carefully described in the context of the existing knowledge. A major revision is needed prior to further consideration.

Reviewer #2 (Remarks to the Author):

The authors proposed using biologically informed neural networks (BINN) to identify the degrees of severity in two proteomics datasets of patients with AKI and COVID-19. The authors showed that the developed BINN models outperformed other models in predicting severity based on proteome profiles. The authors discussed different aspects of the model implementation, evaluated its performance, and studied proteins, pathways, and processes that explain the predicted outcomes. The authors compared

the nominated biomarkers by BINN to markers identified using standard differential expression analysis. The authors showed that BINN could capture biological signals that standard methods may overlook. The authors reported the results and described the methods in a highly organized manner and well-written manuscript. The authors discussed the proposed model limitations and proposed potential enhancements for future development.

The reviewer, however, raises the following points that may enhance the presentation and solidify the results,

- The title is too generic. The reader should be able to identify the context of the manuscript from the title.
- Equation 2: It is not clear how the subgraph around a particular node 'n' is calculated
- Figure 1: The BINN graph shows a fully connected network while it should be sparse
- Figure 2 is hard to interpret with the overlapping confidence interval. It may be helpful to describe the meaning of the shade around the AUC means.
- k-fold is mentioned multiple times without a specific number for k
- Figure 2: dashed line from random performance is at 0.5 however, given that the data is class-imbalanced, the random performance should be different than 0.5
- Figure 3: it is not clear why larger nodes have a lighter color. For example, H4 is bigger in size compared to APOA4 but it is lighter in color.
- Figure 4: the authors mention, "The lower panel shows which fraction of samples identified the given protein." What does "identify" mean here? More explanation may be helpful here
- Figure 5: How would the reader interpret the colors in the graph?
- Figure 7: Panel c: how is it possible that the number of nodes decreases with increasing the number of layers? Does the y-axis represent the total number of nodes? or the nodes per layer?
- Figure 10 matrices are symmetric. It may be helpful to remove the lower half of the matrix.
- It may be helpful to discuss the clinical use cases of the proposed model.

Reviewer #3 (Remarks to the Author):

The manuscript "Interpreting biologically informed neural networks for enhanced biomarker discovery and pathway analysis" by Hartman and colleagues describes the use of NN for novel pathway and biomarker identification using sepsis and COVID MS bottom-up data. The work is very thoroughly done and logically organized. The pipeline developed can sig. help the proteomics community to find robust markers and their associated pathways. Despite the good presentation, there are a few critical points that still need to be addressed.

1. one of the major problems with bottom-up proteomics is the identification of proteins with only two or three peptides, resulting in a high proportion of protein groups. This has a direct impact on the analysis of metabolic pathways, since the primary protein name is usually taken, often only in alphabetical order. The authors have taken great effort designing the pipeline, but have not addressed this critical phenomenon, or it is not apparent to the present reviewer. The high proportion of protein groups has a direct impact not only on the marker/pathway definition, but also on the background matrix and thus also on the calculated p values. This part needs to be described and discussed more clearly, including which background matrix was used. In addition, some iterations (bootstrapping) should be included, such as inserting independently the second, third hit... of the protein group and then comparing their pathway p-values. If this is already somehow incorporated into the NN, it should be highlighted and described in depth as it would be an important insight.

2. it would be very interesting to see how the presented analysis methods would look

with non-MS data such as Olink and whether they are consistent with bottom-up ARDS data, studies or cohort-wide. For example, in the case of Olink, the protein group phenomenon would be obsolete, and the data could be used to validate the MS data-driven pathways from the present study. To my knowledge, several of these omics analyses of COVID-19 and bacterial sepsis-induced ARDS are also publicly available.

3. I am somewhat puzzled by the statement that "Several of the top-ranking proteins in the sepsis-BINN were known biomarkers for inflammation and have been documented to be altered during severe sepsis, such as CD14 [33, 34], FA10 [35], H4 [36], and OSTP [37], however, proteins related to metabolic processes, such as apolipoproteins (APOB, APOA1, APOA2 and APOA4) were also identified. Notably, these were not included in the top-ranking proteins by DE-score and wouldn't be identified with classical differential expression analysis."

There are several papers showing that apolipoproteins are sig. down-regulated in severe sepsis using classical analysis. For example, Michalik et. al. described the down-regulation of APOLIPO proteins 2020 in mSystems using DIA.

4. It is not a must to pass the review process, but it would be another plus if the described analysis pipeline was offered as a Shiny app. This would automatically lead to increased usage of the well-designed tool.

Reviewer #1 (Remarks to the Author):

In this manuscript, the authors implemented an explainable artificial intelligence (XAI) framework of biologically informed neural networks (BINN), by mirroring the complex molecular relationships curated in the Reactome pathway database as the connections of neurons. Using this BINN, they compiled two plasma proteomic data sets for two immune-related diseases, sepsis induced acute kidney injury (AKI) and COVID-19 disease. Using the n-fold cross-validation, the classification accuracy of BINN was compared to other existing method, with much higher AUC and AUPR values. Two major flaws existed in the current form, which must be carefully addressed prior to further consideration.

1. The technical aspect. The authors claimed that AUC and AUPR values of BINN reached to 0.99 and 0.99 for septic-AKI, and 0.95 and 0.96 for COVID-19 disease. The authors must exclude the possibility of over-fitting. Unlike genomic or transcriptomic profiling, the fluctuation of proteomic quantification is highly dynamic. For example, if a patient just drinks a small cup of waters, the plasma proteomic quantification might undergo a considerable change. And due to a biological process, circadian rhythm, the proteomic compositions will be significantly fluctuated in morning and night for a single patient. Also, the coverage of proteomic profiling is quite low, and many proteins detectable in some experiments will be missed in other runs. So imputation is needed to complement the data, and different imputation methods influence the identification of protein biomarkers. Third, the genetic background of people worldwide is not identical, and proteomic compositions are obviously different in people. So for a highly dynamic and unintegrated data type, how a machine learning approach can reach such a high accuracy? The only explanation is that the computational model is highly over-fitted. To evaluate the robustness of BINN, the n-fold cross-validation is not enough, and the authors must collect published cohorts published by other groups. Such published data in either sepsis or COVID-19 are very easy to be obtained, and these data should be used as independent data set not used for training. In addition, the authors should collect an enough number of proteomic data from healthy volunteers as the control, not only just classification of mild or severe disease.

The reviewer raises an important concern regarding the possibility of over-fitting. We would like to point out that there are potentially other reasons why the machine learning models can reach high accuracy for these samples. Previous work has shown that infectious diseases such as sepsis and COVID-19 introduce a drastic change in the blood plasma proteome (see Karlsson et al, 2018, *Mol. Cell Proteomics*, and Fisher et al. 2021, *Mol. Cell Proteomics* for examples) compared to healthy controls and to other diseases. It is our experience that factors e.g. lifestyle factors and biological processes such as circadian rhythm, typically introduce smaller changes in the blood plasma proteome compared to sepsis/Covid-19. We speculate that this may be one explanation why we and others can show that machine learning models can achieve high AUC/AUPR. See the following articles for examples:

- Palma Medina, L.M., Babačić, H., Dzidic, M. *et al.* Targeted plasma proteomics reveals signatures discriminating COVID-19 from sepsis with pneumonia. *Respir Res* 24, 62 (2023). <https://doi.org/10.1186/s12931-023-02364-y>
- Demichev, V., Tober-Lau, P., Lemke, O., Nazarenko, T., Thibeault, C., Whitwell, H., Röhl, A., Freiwald, A., Szyrwiel, L., Ludwig, D., Correia-Melo, C., Aulakh, S. K., Helbig, E. T.,

Stubbemann, P., Lippert, L. J., Grüning, N. M., Blyuss, O., Vernardis, S., White, M., Messner, C. B., ... Kurth, F. (2021). A time-resolved proteomic and prognostic map of COVID-19. *Cell systems*, 12(8), 780–794.e7.
<https://doi.org/10.1016/j.cels.2021.05.005>

To address the comment of overfitting, we employed several techniques to minimize overfitting during training, such as dropout, batch normalization and an L2-penalty. We have in the revised version added the loss curves as Supplementary figure 13. These curves show that the models are not over-fitting, as the training loss is equal or higher than the validation loss. As suggested by the reviewer, we have furthermore included a test set of 99 samples for COVID as an independent data set. These samples were generated at a different hospital than the ones used for the training data, and analyzed using the same pipeline. We were not able to find a suitable test set for sepsis. The trained model achieved an accuracy of 86.8% on the previously unseen testing data, which slightly surpasses the accuracy reached on the training data. Taken together, these results demonstrate that our models are not over-fitting on the training data, and that the plasma proteomic content in sepsis and COVID-19 is sufficient to reach the accuracies regarding AUCs and AUPRs. We have included the new dataset, added a new figure, and added new paragraphs to the methods and results sections in the revised version of the manuscript as follows:

“To ensure that the measures taken to minimize the risk of overfitting such as the use of dropout, batch normalization and L2-regularization, were effective, the COVID-model was tested on an independent cohort consisting of 99 samples. The model achieved an accuracy of 87% on the testing-cohort, confirming that it generalizes to unseen data and that overfitting did not occur. This is further motivated by the loss-curves, as the evaluated loss during training and validation are matched (figure 13).”

2. The biological aspect. A predictor with a nearly 100% accuracy means that all factors, including genetic differences, life styles, and geographical features, as well as real proteomic changes, were almost fully captured. The authors should explain what they observed from SHAP interpretation, beyond the proteomic signatures. Also, due to the much higher performance of BINN, it can be expected that new biomarkers can be identified from the sample classification. Without additional experimental validations is OK, however, the authors should put their findings in the context of the literature, to demonstrate why the newly predicted biomarkers are truly involved in septic-AKI or COVID-19.

This comment made us realize that the first version of the manuscript was not clear. The AUC values for the BINN are 0.99 and 0.95 for the sepsis and covid BINNS, but the overall accuracies (correct classifications / total classifications) of the COVID (~87%) and sepsis (~98%) BINNs show they are not perfect classifiers. The high AUCs indicate that both BINNs can distinguish between true positives and true negatives accurately across varying classification thresholds, but this is not the same thing as directly measuring accuracy, as accuracy is the correct classifications over total classifications at a fixed threshold. We have made the manuscript more clear by including the accuracy in the text under the “Method comparison” section of the Results:

“Evaluation was performed on the complete datasets using k-fold cross-validation ($k = 3$). All machine learning methods achieved AUC-scores of >0.75 , but the BINNs resulted in the best performances (ROC-AUC 1: 0.99 ± 0.00 and 0.95 ± 0.01 , PR-AUC 2: 0.99 ± 0.00 and 0.96 ± 0.01) on the septic AKI and COVID-dataset respectively (figure 2A,B,C,D). Both BINNs achieved a high true positive and true negative rate (sepsis: $94 \pm 2\%$, $100 \pm 0\%$, COVID: $87 \pm 2\%$, $92 \pm 1\%$) (figure 2E,F). The total accuracy for the models were: $98.6 \pm 2\%$ (septic AKI) and $87.5 \pm 3\%$ (COVID-19). Additionally, both the sepsis and the COVID-BINN attained the highest precision and recall-rates out of all methods, achieving a precision of: 0.99 ± 0.020 , 0.87 ± 0.011 , and recall of: 1.0 ± 0.0 , 0.88 ± 0.022 respectively.”

Beyond the proteomic signatures, we utilized SHAP to analyze the importance of the biological pathways and processes used to construct the BINNs. This allows us to intelligently interrogate pathways and potentially uncover some insight into the phenotypes related to the proteome (Figure 3, Figure 5 and Figure 6).

As pointed out by the reviewer, performance of BINN enables the identification of new biomarkers. To put these potential protein markers into context we have added references and text to the result section and the discussion section of the manuscript to demonstrate that the newly predicted biomarkers are of relevance for septic-AKI or COVID-19.

“Several of the top-ranking proteins in the sepsis-BINN were known biomarkers for inflammation and have been documented to be altered during severe sepsis, such as CD14 [35, 36], FA10 [37], H4 [38], and OSTP [39]. For example, soluble CD14 has previously been shown to be a promising and rapid responding candidate diagnostic marker for neonatal early and late onset sepsis [40]. Additionally, proteins related to metabolic processes, such as apolipoproteins (APOB, APOA1, APOA2 and APOA4) which also undergo alterations during sepsis [41, 34], were identified.”

And

“Similarly, many of the most important proteins in the COVID-BINN have previously been proposed as biomarkers for the distinction between moderately and critically ill COVID-patients, such as GELS, ZA2G [43] and S10A8 [44].”

The section on biomarkers in the discussion has also been expanded:

“Biomarker discovery in the context of BINNs is performed by calculating the feature importance of the initial layer of the network. Several of the most important proteins in the sepsis and COVID-BINNs were known biomarkers of the respective disease, however, they differed from the most differentially expressed proteins. Important proteins were not necessarily the most significantly differentially expressed (supplementary 10). Proteins highlighted in the sepsis-BINN were both prognostic inflammatory biomarkers such as CD14, FA10 and OSTP, but also biomarkers related to metabolic proteins such as apolipoproteins. Apolipoproteins have been found to correlate to 30-day mortality in sepsis, as well as platelet activation and monocyte activation affecting patient outcomes [44] and were not among the top proteins by differential expression. The COVID-BINN highlighted several proteins which have been proposed as diagnostic markers for criti-

cally ill COVID-patients, such as GELS, ZA2G and SA108 [45, 49, 42]. Notably, levels of gelsolin and S10A8 in plasma and have been found to be directly correlated to the severity of COVID-19 [49, 43]. Biomarker identification with BINNs and classical differential expression may be seen as complementary approaches, and both may provide value to an analysis. Whereas differential expression analysis is guaranteed to provide proteins with a high fold change and low p-value, as this is the selection criteria, a BINN will provide the proteins which are important in a classification context when taking biological processes into account.”

Taken together, my overall thoughts are: 1) The BINN framework is interesting and potentially used for sample classification and biomarker finding, in a higher interpretable manner; 2) Additional cohorts not used for training should be compiled from published literature for a justified testing and comparison. 3) Newly uncovered protein biomarkers should be carefully described in the context of the existing knowledge. A major revision is needed prior to further consideration.

We have included an additional cohort to demonstrate that the model is not overfitting. In addition, we have included another proteomics data set generated using the Olink platform to demonstrate that BINNs generalizes to other types of proteomics data types. Lastly, we have discussed the uncovered protein biomarkers in context of the existing knowledge.

Reviewer #2 (Remarks to the Author):

The authors proposed using biologically informed neural networks (BINN) to identify the degrees of severity in two proteomics datasets of patients with AKI and COVID-19. The authors showed that the developed BINN models outperformed other models in predicting severity based on proteome profiles. The authors discussed different aspects of the model implementation, evaluated its performance, and studied proteins, pathways, and processes that explain the predicted outcomes. The authors compared the nominated biomarkers by BINN to markers identified using standard differential expression analysis. The authors showed that BINN could capture biological signals that standard methods may overlook. The authors reported the results and described the methods in a highly organized manner and well-written manuscript. The authors discussed the proposed model limitations and proposed potential enhancements for future development.

The reviewer, however, raises the following points that may enhance the presentation and solidify the results,

- The title is too generic. The reader should be able to identify the context of the manuscript from the title.

We are hesitant to change the title as BINN is generic and can be applied to many different data types and biological/clinical applications. In the first version of the manuscript, we showcase the BINNs using two mass spectrometry data sets in infectious disease. The implementation is however agnostic to the underlying graph and inputs used for the creation of the network. To demonstrate this more clearly, we included another showcase based on a data set generated with the Olink platform. Furthermore, we are in ongoing work using BINN

for several other applications to detect protein and pathway-changes in microbes, cells and tissues. This makes us confident that BINN can be used on several different types of data in several disparate biological settings and that a more general title is fitting for the manuscript.

- Equation 2: It is not clear how the subgraph around a particular node 'n' is calculated

This has been clarified in the text: "The complete subgraph of node n is defined as the complete set of predecessors and successors originating from n in the directed graph G "

- Figure 1: The BINN graph shows a fully connected network while it should be sparse

The figure has been corrected to show a sparse network.

- Figure 2 is hard to interpret with the overlapping confidence interval. It may be helpful to describe the meaning of the shade around the AUC means.

It has been clarified that the bands showcase the 95% confidence intervals in the figure legend.

- k-fold is mentioned multiple times without a specific number for k

It has been clarified that k-fold cross validation was performed with 3 folds when mentioned in the text.

- Figure 2: dashed line from random performance is at 0.5 however, given that the data is class-imbalanced, the random performance should be different than 0.5

The diagonal line has been removed from the ROC-plots in figure 2.

- Figure 3: it is not clear why larger nodes have a lighter color. For example, H4 is bigger in size compared to APOA4 but it is lighter in color.

The size of the nodes are defined by the sum of the incoming and outgoing values, and therefore incorporate both the importance and the number of incoming and outgoing connections. H4 has more outgoing edges than APOA4 and is therefore rendered as larger, even though APOA4 is more important. The color of the node is directly proportional to the importance. This has been clarified with the following text in the figure legend: "Node sizes are proportional to the sum of incoming and outgoing values, and therefore take the connectivity and importance into account."

- Figure 4: the authors mention, "The lower panel shows which fraction of samples identified the given protein." What does "identify" mean here? More explanation may be helpful here

The proteomic data is not complete as not all proteins were identified in all samples. In cases where a protein is not identified in a sample, a value is imputed (we impute with 0s). We made this more clear in the revised version of the manuscript by adding the following text in the figure legend: "The lower panel shows in which fraction of the samples a given

protein was identified. When training the predictors, proteins which weren't identified in a sample were imputed with a 0.”

- Figure 5: How would the reader interpret the colors in the graph?

Colorbars have been added to clarify that the colours correlate with relative importance. These were also added to figure 3.

- Figure 7: Panel c: how is it possible that the number of nodes decreases with increasing the number of layers? Does the y-axis represent the total number of nodes? or the nodes per layer?

The y-label has been altered to reflect that the nodes per layer is plotted, not the total number of nodes.

- Figure 10 matrices are symmetric. It may be helpful to remove the lower half of the matrix.

The lower parts of the correlation-matrices have been removed.

- It may be helpful to discuss the clinical use cases of the proposed model.

This is an interesting idea. It is possible that BINN could potentially contribute to the development of clinical use cases such as biomarker discovery, drug target discovery and classification of subphenotypes. However, we prefer to refrain from an in-depth discussion of clinical use cases in the manuscript as BINN is in its current form primarily intended as a research tool for preclinical and translational studies. We have mentioned several examples of clinical applications in the discussion section:

“In addition, the BINN-package can be used to analyze data from different platforms. The ability for BINNs to analyze data from different proteomics platforms and with different types of underlying graphs extends its reach in biomedical and clinical applications including biomarker discovery, drug target discovery and subphenotype classifications as these problems are highly multifaceted. ”

Reviewer #3 (Remarks to the Author):

The manuscript "Interpreting biologically informed neural networks for enhanced biomarker discovery and pathway analysis" by Hartman and colleagues describes the use of NN for novel pathway and biomarker identification using sepsis and COVID MS bottom-up data. The work is very thoroughly done and logically organized. The pipeline developed can sig. help the proteomics community to find robust markers and their associated pathways. Despite the good presentation, there are a few critical points that still need to be addressed.

1. one of the major problems with bottom-up proteomics is the identification of proteins with only two or three peptides, resulting in a high proportion of protein groups. This has a direct impact on the analysis of metabolic pathways, since the primary protein name is usually taken, often only in alphabetical order. The authors have taken great effort designing the

pipeline, but have not addressed this critical phenomenon, or it is not apparent to the present reviewer. The high proportion of protein groups has a direct impact not only on the marker/pathway definition, but also on the background matrix and thus also on the calculated p values. This part needs to be described and discussed more clearly, including which background matrix was used. In addition, some iterations (bootstrapping) should be included, such as inserting independently the second, third hit... of the protein group and then comparing their pathway p-values. If this is already somehow incorporated into the NN, it should be highlighted and described in depth as it would be an important insight.

We completely agree with this comment and the comment made us realize that some critical information was missing in the previous version of the manuscript. To minimize the risk of incorrect assignment, and therefore biased downstream analysis, we have restricted the proteome maps in this study to only include proteotypic peptides. In this way, we know that each measured peptide only maps to 1 protein, so that each protein group is singular in nature (ie. each peptide can only belong to 1 protein). This makes protein quantification and the downstream analysis of pathways more straightforward. Since this was unclear in the text, we have emphasized this in the Results and the Methods in the following manner:

In the section “Construction of biologically informed neural networks” of the Results:

“Proteins for both the septic AKI and COVID-19 datasets were quantified using proteotypic peptides from the mass-spectrometry based proteome maps to ensure unique protein group membership for downstream analysis.”

In the section “Data processing” of the Results:

“Samples were then mean-normalized to remove any bias in the data and proteins were quantified from proteotypic peptides using a python implementation of the relative quantification iq algorithm [54]”

2. it would be very interesting to see how the presented analysis methods would look with non-MS data such as Olink and whether they are consistent with bottom-up ARDS data, studies or cohort-wide. For example, in the case of Olink, the protein group phenomenon would be obsolete, and the data could be used to validate the MS data-driven pathways from the present study. To my knowledge, several of these omics analyses of COVID-19 and bacterial sepsis-induced ARDS are also publicly available.

Demonstrating that BINN works with different types of data is certainly a good addition to the paper. We have now downloaded and analyzed the dataset from the study: *“Urine-based multi-omic comparative analysis of COVID-19 and bacterial sepsis-induced ARDS”* Batra et al. 2023, where they analyzed the proteomic content of urine from patients suffering from bacterially induced septic ARDS (N=17) and ARDS induced by COVID-19 (N=42) as well as healthy controls (N=25) using Olink. We decided to create a multi-output BINN to classify these three classes. The resulting BINN can now be seen in figure 6.

The following section was added to results:

“Cross-platform generalizability

To demonstrate the ability of the BINN to generalize cross-platform, a proteomics dataset generated using the Olink-platform (Uppsala, Sweden) was analyzed. Here, the proteomic content of urine from patients suffering from bacterial sepsis-induced acute respiratory distress syndrome (ARDS) (17 samples) and COVID-19-induced ARDS (42 samples) were analyzed. A cohort of healthy controls was also included (25 samples). The pre-processed data data was downloaded and analyzed without modifications.

The Olink-BINN was trained to classify between the three classes: COVID-19-induced ARDS, bacterial sepsis-induced ARDS and healthy controls, and was evaluated using k-fold cross validation ($k = 3$). This is a three-class classification problem with a low number of samples, however, the Olink-BINN still managed to identify healthy and COVID-19-induced ARDS with high accuracy (true positive rates: healthy: 0.8 ± 0.12 , COVID-19-ARDS: 0.81 ± 0.03). The low number of samples and the heterogeneity of the bacterial sepsis-induced ARDS resulted in a low true positive rate for this class (0.29 ± 0.15) (figure 6). Additionally, the BINN highlighted several pathways with relevance to ARDS such as the G-coupled protein receptors-pathway [49], which contributed to the Signal Transduction-pathway as being one of the most important (figure 6)."

The following section was added to the discussion:

"In addition, the BINN-package can be used to analyze data from different platforms. The ability for BINNs to analyze data from different proteomics platforms and with different types of underlying graphs extends its reach in biomedical and clinical applications including biomarker discovery, drug target discovery and subphenotype classifications as these problems are highly multifaceted."

Methods have also been adjusted to include the Olink data.

3. I am somewhat puzzled by the statement that "Several of the top-ranking proteins in the sepsis-BINN were known biomarkers for inflammation and have been documented to be altered during severe sepsis, such as CD14 [33, 34], FA10 [35], H4 [36], and OSTP [37], however, proteins related to metabolic processes, such as apolipoproteins (APOB, APOA1, APOA2 and APOA4) were also identified. Notably, these were not included in the top-ranking proteins by DE-score and wouldn't be identified with classical differential expression analysis."

There are several papers showing that apolipoproteins are sig. down-regulated in severe sepsis using classical analysis. For example, Michalik et. al. described the down-regulation of APOLIPO proteins 2020 in mSystems using DIA.

Thank you for pointing this out. The text in the previous version of the manuscript was not clear. The text was intended to highlight that the apolipoproteins do not place within the top ranking proteins by DE-score, but they were highlighted in the sepsis-BINN. We have altered this section to reflect this, and have included Michalik et al. 2020 as a reference:

"Several of the top-ranking proteins in the sepsis-BINN were known biomarkers for inflammation and have been documented to be altered during severe sepsis, such as

CD14 [35, 36], FA10 [37], H4 [38], and OSTP [39]. For example, soluble CD14 has previously been shown to be a promising and rapid responding candidate diagnostic marker for neonatal early and late onset sepsis [40]. Additionally, proteins related to metabolic processes, such as apolipoproteins (APOB, APOA1, APOA2 and APOA4) which also undergo alterations during sepsis [41, 34], were identified. “

4. It is not a must to pass the review process, but it would be another plus if the described analysis pipeline was offered as a Shiny app. This would automatically lead to increased usage of the well-designed tool.

We thank the reviewer for the suggestion. Although we decided to not implement BINN as a Shiny application, we have written a thorough documentation for the BINN python-package, including an API reference and examples. The documentation is available here: <https://infectionmedicineproteomics.github.io/BINN/>. Additionally, it now supports an install via pip. We will consider adding a GUI for future releases.

Reviewer #1 (Remarks to the Author):

In revision, the authors considerably improved the manuscript, but did not very carefully address my concerns. My major points are below:

1. The authors stated that "...the COVID model was tested on an independent cohort consisting of 99 samples." This data set was not referenced, and I do not know where and how the authors obtained this data set.

2. The authors stated that "We were not able to find a suitable test set for sepsis". A simple search of PubMed returned some papers, e.g., "Targeted plasma proteomics reveals signatures discriminating COVID-19 from sepsis with pneumonia" (Respir Res . 2023 Feb 24;24(1):62. <https://pubmed.ncbi.nlm.nih.gov/36829233/>). In this paper, 276 plasma proteins involved in Inflammation, organ damage, immune response and coagulation were measured in healthy controls, COVID-19 patients during acute and convalescence phase, and sepsis patients. A more careful literature search should be conducted.

3. I suggest that "the authors must collect published cohorts published by other groups". The number of data sets cannot be equal to only 1. The authors should try their best to collect various independent data sets as more as possible to evaluate the performance of BINN, because the superiority of BINN was not very convincing based on the currently limited tests.

Reviewer #2 (Remarks to the Author):

The authors provided an updated version of the manuscript with more experiments, updated figures, clarified descriptions, and a better presentation. The reviewer is still concerned that the title is too generic and does not genuinely reflect the context of the work reported in this manuscript. For example, it is unclear what the word "biomarkers" in the title refers to; is it genomics, transcriptomic, or proteomics marker?

Reviewer #3 (Remarks to the Author):

The authors have addressed all points and adapted the manuscript accordingly. I would have been happy with a shiny app, as it would bring more users to BINN, but I also agree with the published package, which is relatively user-friendly, but requires some degree knowledge. However, from my point of view the manuscript is now ready for publication.

REVIEWER COMMENTS

Reviewer #1 (Remarks to the Author):

In revision, the authors considerably improved the manuscript, but did not very carefully address my concerns. My major points are below:

1. The authors stated that "...the COVID model was tested on an independent cohort consisting of 99 samples." This data set was not referenced, and I do not know where and how the authors obtained this data set.

We have now clarified where the data was obtained from in both the methods and results sections:

Methods:

"The COVID-19 dataset consisting of the raw data matrix of quantified precursors and design matrix with patient annotations were downloaded from PRIDE (PXD025752) and re-analyzed. The original study reports two cohorts from different hospitals whereof the samples gathered at Charité containing 687 samples was used for training, and the samples gathered at Innsbruck consisting of 99 samples were used as a testing-cohort. These are available under the same PRIDE identifier."

Results:

"To ensure that the measures taken to minimize the risk of overfitting such as the use of dropout, batch normalization and L2-regularization, were effective, both the COVID and septic AKI-models were tested on independent cohorts. The COVID-BINN was tested on a cohort consisting of 99 samples. These were reported in the same study as the samples comprising the training-set, but were gathered at a different hospital [29]. The AKI-BINN on a cohort consisting of 56 samples. These samples were collected in the FINNAKI-study [28], but has not been published previously. The COVID-BINN and sepsis-BINN achieved accuracies of 87% and 91% respectively on the testing-cohorts, confirming that they generalize to unseen data and that overfitting did not occur. This is further motivated by the loss-curves, as the evaluated loss during training and validation are matched (figure 13)."

2. The authors stated that "We were not able to find a suitable test set for sepsis". A simple search of PubMed returned some papers, e.g., "Targeted plasma proteomics reveals signatures discriminating COVID-19 from sepsis with pneumonia" (Respir Res . 2023 Feb 24;24(1):62. <https://pubmed.ncbi.nlm.nih.gov/36829233/>). In this paper, 276 plasma proteins involved in Inflammation, organ damage, immune response and coagulation were measured in healthy controls, COVID-19 patients during acute and convalescence phase, and sepsis patients. A more careful literature search should be conducted.

We agree with the reviewer that there are many publications related to proteomics in sepsis overall, however, the BINN is trained to distinguish specifically between two different subphenotypes of septic Acute Kidney Injury (AKI). AKI is a specific disease that is most commonly related to sepsis, but is distinctly different from other diseases, such as sepsis

with pneumonia. For that reason, we unfortunately cannot use the suggested dataset, as it does not measure the specific septic-AKI proteome. We did conduct a thorough literature search, but to our knowledge, there is no acceptable published cohort of septic AKI using mass spectrometry to stratify these 2 previously defined subphenotypes from blood plasma. We realize that our naming of the associated dataset and BINN model relating to septic AKI may have been confusing, so we have renamed the sepsis-BINN to AKI-BINN and make sure to refer to the dataset as the septic AKI dataset where appropriate.

However, to meet the reviewers request we processed 56 new unpublished samples originating from the FINNAKI study to test how the AKI-BINN will perform on previously unseen data and to provide extra checks for overfitting. These samples were processed using the same protocol as the previously published samples to generate a test-set for the AKI-BINN. On this test set, the BINN achieved an accuracy of 91%, demonstrating that the AKI-BINN generalizes to previously unseen data. The protocols for the analysis of these new samples have been added to methods, and adjustments have been made throughout the paper to reflect that the new dataset has been added.

Methods:

“To generate a testing-dataset for the sepsis-model, 56 previously unpublished samples from the FINNAKI [28] study were processed. The protocol for sample preparation and data acquisition is identical to how the previously published septic AKI dataset was generated by Scott et al. [55] Briefly, samples were processed using the Agilent AssayMAP Bravo Platform (Agilent Technologies, Inc.) per manufacturer's protocol. The resulting samples were spiked with synthetic iRT peptides (JPT Peptide Technologies, GmbH, Berlin, Germany) before liquid chromatography-mass spectrometry analysis. The raw mass spectrometry files were processed in the same manner as the previously published septic AKI-samples using the same spectral library, as per above.”

We have also updated the Results section to include text relating to the new analysis of the COVID and AKI test data.

Results:

“To ensure that the measures taken to minimize the risk of overfitting such as the use of dropout, batch normalization and L2-regularization, were effective, both the COVID And septic AKI-models were tested on independent cohorts. The COVID-BINN was tested on a cohort consisting of 99 samples. These were reported in the same study as the samples comprising the training-set, but were gathered at a different hospital [29]. The AKI-BINN on a cohort consisting of 56 samples. These samples were collected in the FINNAKI-study [28], but has not been published previously. The COVID-BINN and AKI-BINN achieved accuracies of 87% and 91% respectively on the testing-cohorts, confirming that they generalize to unseen data and that overfitting did not occur. This is further motivated by the loss-curves, as the evaluated loss during training and validation are matched (figure 13).”

3. I suggest that “the authors must collected published cohorts published by other groups”. The number of data sets cannot be equal to only 1. The authors should try their best to collect various independent data sets as more as possible to evaluate the performance of

BINN, because the superiority of BINN was not very convincing based on the currently limited tests.

We have now added a test set for the sepsis-BINN, and we therefore now include 5 sets of data in the study, whereof 3 are for training/validation and 2 for testing. Below is a summary of the data included in the study:

Model	# samples	Origin	training/validation/testing
sepsis-BINN	141	FINNAKI study, previously published	training/validation
sepsis-BINN	56	FINNAKI study, published in our study	testing
covid-BINN	687	Charité hospital	training/validation
covid-BINN	99	Innsbrück hospital	testing
olink-BINN	84	Weill Cornell Biobank of Critical Illness	training/validation

Further, we've added a discussion on the generalizability of the BINNs to the discussion:

Discussion

“The ability of a model to generalize to new data depends on the quality, diversity, and size of training datasets to capture the underlying distributions of the data. Adequate dataset size can help prevent overfitting, and provide coverage of various scenarios to facilitate real-world applicability. To evaluate how BINNs generalize to new data, we provide previously unseen test sets for both the AKI and COVID-BINNs. The high accuracies of both the AKI and COVID-BINNs (91% and 87% respectively) suggest their ability to generalize to unseen samples effectively. However, since the number of samples in biological experiments are typically decided based on availability, the training sets used in our study may not fully represent the complete underlying distributions, and could be expanded to maximize the potential of BINNs based on the experiment. “

Reviewer #2 (Remarks to the Author):

The authors provided an updated version of the manuscript with more experiments, updated figures, clarified descriptions, and a better presentation. The reviewer is still concerned that the title is too generic and does not genuinely reflect the context of the work reported in this manuscript. For example, it is unclear what the word "biomarkers" in the title refers to; is it genomics, transcriptomic, or proteomics marker?

We agree that this should be clarified and have changed the title to: *“Interpreting biologically informed neural networks for proteomic biomarker discovery and pathway analysis”*

Reviewer #3 (Remarks to the Author):

The authors have addressed all points and adapted the manuscript accordingly. I would have been happy with a shiny app, as it would bring more users to BINN, but I also agree with the published package, which is relatively user-friendly, but requires some degree knowledge. However, from my point of view the manuscript is now ready for publication.

We thank the reviewer for the feedback which greatly increased the quality of the manuscript.

Reviewer #1 (Remarks to the Author):

The authors carefully addressed all my concerns, and included new data sets not used for training to test the performance of BINN. I do not have any additional comments, and agree that the manuscript can be accepted in the current form.

REVIEWERS' COMMENTS

Reviewer #1 (Remarks to the Author):

The authors carefully addressed all my concerns, and included new data sets not used for training to test the performance of BINN. I do not have any additional comments, and agree that the manuscript can be accepted in the current form.

We thank the reviewer for great feedback and a productive review-process which has greatly improved the manuscript.